# Dorsal root ganglion macrophages contribute to both the initiation and persistence of neuropathic pain

Xiaobing Yu [1]*, Hongju Liu[1,2], Katherine A. Hamel [3], Maelig G. Morvan [4], Stephen Yu[3], Jacqueline Leff[1], Zhonghui Guan[1], Joao M. Braz [3] & Allan I. Basbaum[3]*

Paralleling the activation of dorsal horn microglia after peripheral nerve injury is a significant expansion and proliferation of macrophages around injured sensory neurons in dorsal root ganglia (DRG). Here we demonstrate a critical contribution of DRG macrophages, but not those at the nerve injury site, to both the initiation and maintenance of the mechanical hypersensitivity that characterizes the neuropathic pain phenotype. In contrast to the reported sexual dimorphism in the microglial contribution to neuropathic pain, depletion of DRG macrophages reduces nerve injury-induced mechanical hypersensitivity and expansion of DRG macrophages in both male and female mice. However, fewer macrophages are induced in the female mice and deletion of colony-stimulating factor 1 from sensory neurons, which prevents nerve injury-induced microglial activation and proliferation, only reduces macrophage expansion in male mice. Finally, we demonstrate molecular cross-talk between axotomized sensory neurons and macrophages, revealing potential peripheral DRG targets for neuropathic pain management.

[1] Department of Anesthesia and Perioperative Care, University of California San Francisco, San Francisco, California, USA. [2] Department of Anesthesiology, Peking Union Medical College Hospital, Beijing, China. [3] Department of Anatomy, University of California San Francisco, San Francisco, California, USA. [4] Department of Medicine, University of California San Francisco, San Francisco, California, USA. *email: Xiaobing.Yu@ucsf.edu; Allan.Basbaum@ucsf.edu

There is now considerable evidence that molecular and cellular interactions among spinal dorsal horn neurons and microglia, the resident macrophages of the central nervous system (CNS), are important contributors to the induction and maintenance of neuropathic pain following peripheral nerve injury[1,2]. Paralleling the activation of dorsal horn microglia after peripheral nerve injury, several recent studies demonstrated that there is also a significant increase in the number of macrophages in dorsal root ganglia (DRG) ipsilateral to the injury[3,4], suggesting that pro-nociceptive neuronal and non-neuronal cellular interactions also occur in the DRG. Unclear, however, is the extent to which the increase of DRG macrophages contributes to the pain phenotype induced by nerve injury.

To address this question, pharmacological and genetic approaches have been used to examine the behavioral consequence of depleting DRG macrophages in a nerve injury setting. For example, Cobos et al.[5] reported that clodronate-mediated killing of DRG macrophages reduced the mechanical allodynia induced by nerve injury. However, other groups could not confirm those findings[6–9], conceivably due to variability in the efficacy of clodronate[5,6]. Other studies demonstrated clodronate killing of macrophages in injured peripheral nerves and concluded that these macrophages were critical to the neuropathic pain development[10,11]. However, as these studies did not examine the DRG, the contribution of DRG macrophages could not be ruled out.

Selective CSF1 receptor (CSF1R) antagonists have also been used to induce cell death, by blocking CSF1 signaling to macrophages. However, as these CSF1R antagonists readily cross the blood–brain barrier (BBB), they concurrently deplete both CNS microglia[12–14] and macrophages[13]. As a result, this approach cannot address selectively the contribution of DRG macrophages. The same limitation confounds the interpretation of results from studies that used transgenic mouse lines that express a drug-inducible suicide gene, e.g., herpes simplex virus type 1 thymidine kinase (*CD11b-TK*)[15] or diphtheria toxin receptor (*CD11b-DTR*[16], *LysM-DTR*[17], and *Cx3cr1-DTR*[18]) in both microglia and macrophages. As the drugs that induce killing readily cross the BBB, it is impossible to distinguish between the contribution of DRG macrophages and microglia[18–20].

A recent report[21] took a different approach, using the macrophage Fas-induced apoptosis (MAFIA) transgenic mouse line[22]. As for the TK or DTR mice, this line expresses a suicide gene (Fas) under the control of the CSF1R promoter, which is specifically expressed in macrophages and microglia. In these mice, killing is provoked with an FK-binding protein dimerizer, AP20187 (AP). Based on reports that AP does not cross the BBB in MAFIA mice[22], this approach has been used to kill macrophages selectively. Importantly, as this transgenic line coexpresses green fluorescent protein (GFP) under control of the same promoter, it is possible to monitor the distribution of cells that express the suicide gene and the extent of their depletion. Using the MAFIA mice, Shepherd et al.[21] demonstrated reduced nerve injury-induced mechanical hypersensitivity, a hallmark of the neuropathic pain phenotype, after depletion of circulating monocytes and macrophages at the nerve injury site. Moreover, as these authors found that DRG macrophages were spared, they concluded that peripheral macrophages, but not those in the DRG, are the critical contributors to nerve injury-induced neuropathic pain[21]. However, their conclusion was not consistent with an earlier study reporting that selective depletion of peripheral monocytes/macrophages, while sparing DRG macrophages, had limited impact on neuropathic pain development[6].

Given the significant discrepancies in the literature, here we re-examined the question, again using the MAFIA mice. We administered AP systemically, before or after producing the peripheral nerve injury, so as to examine the contribution of peripheral

macrophages to both the initiation and the maintenance of the mechanical hypersensitivity. We report that depletion of macrophages in the DRG, but not at the peripheral nerve injury site, can both prevent the development of and reverse ongoing nerve injury-induced mechanical hypersensitivity, in both male and female mice. In the context of nerve injury, we also uncovered a reciprocal cellular interaction between DRG macrophages and sensory neurons, one that we suggest is relevant to the DRG macrophage contribution to the neuropathic pain phenotype.

## Results

**Nerve injury induces macrophage expansion in the DRG.** Using a fluorescence-activated cell sorting (FACS) analysis of dissociated DRG cells immunostained for the fractalkine receptor, CX3CR1, a marker that selectively defines peripheral monocytic cells and microglia[6], we first quantified macrophages in the DRG after spared nerve injury (SNI). Figure 1a shows that 4 days after nerve injury (POD4), the CX3CR1$^+$ cell population in the L4 and L5 DRG, ipsilateral to the nerve injury, increased by $2.9 \pm 0.4$-fold compared with the DRG on the contralateral, uninjured side. This ipsilateral expansion of macrophages persisted for at least 4 weeks after the nerve injury (POD28).

With a view to determining the origin of the injury-induced macrophage expansion in the DRG, we monitored expression of the chemokine receptor CCR2, which reportedly marks infiltrating macrophages[23], in a double transgenic CCR2-RFP$^{+/-}$/CSF1R-GFP$^{+/-}$ mouse. As expected, we observed significant numbers of CCR2$^+$ macrophages at the peripheral nerve injury site, compared with the contralateral uninjured sciatic nerve (Supplementary Fig. 1A, B); however, we also found many CCR2$^+$ macrophages in the DRG of uninjured mice (Supplementary Fig. 1C–E), indicating that CCR2 is not a reliable marker of infiltrating macrophages. In a separate experiment, we costained the CX3CR1$^+$ macrophages with a Ki67 antibody to mark proliferating cells[24]. One day after nerve injury (POD1), FACS analysis showed that the percentage of Ki67$^+$CX3CR1$^+$ macrophages in the ipsilateral DRG did not differ from the uninjured contralateral DRG (Fig. 1b). However, at POD4, the percentage of Ki67$^+$CX3CR1$^+$ macrophages in the ipsilateral DRG more than doubled (Fig. 1b and Supplementary Fig. 2A, B). Paralleling this result and consistent with previous immunocytochemical findings[25], FACS analysis showed that proliferating microglia (Ki67$^+$CX3CR1$^+$) in the lumbar cord on POD4 ipsilateral to the nerve injury increased by more than twofold compared with the contralateral side (Supplementary Fig. 2C). Together, we conclude that axotomy-induced macrophage expansion in the DRG involves local proliferation. Although we cannot rule out infiltration, we favor the view that proliferation from resident macrophages predominates.

**DRG macrophages are depleted by systemic AP administration.** To determine whether the increase in DRG macrophages contributes to neuropathic pain development, we chose the MAFIA transgenic line[22], which as noted above, expresses GFP and a drug-inducible Fas suicide gene under control of the CSF1R promoter (Supplementary Fig. 3A). The transgene is expressed in CNS microglia (Supplementary Fig. 3B) and tissue macrophages, including DRG (Supplementary Fig. 4). To deplete monocytic cells in the MAFIA mice, we began our studies with five daily intraperitoneal injections of AP (10 mg kg$^{-1}$), a macrophage-depleting dose used in earlier studies[22]. Supplementary Fig. 5 shows that this dose was without effect on baseline mechanical threshold (S5A) and weight (S5B) in wild-type (WT) mice. By contrast, this regimen resulted in significant weight loss and also increased the baseline mechanical threshold 1 day after the last injection in the MAFIA mice (Supplementary Fig. 6A). As these effects clearly complicated our

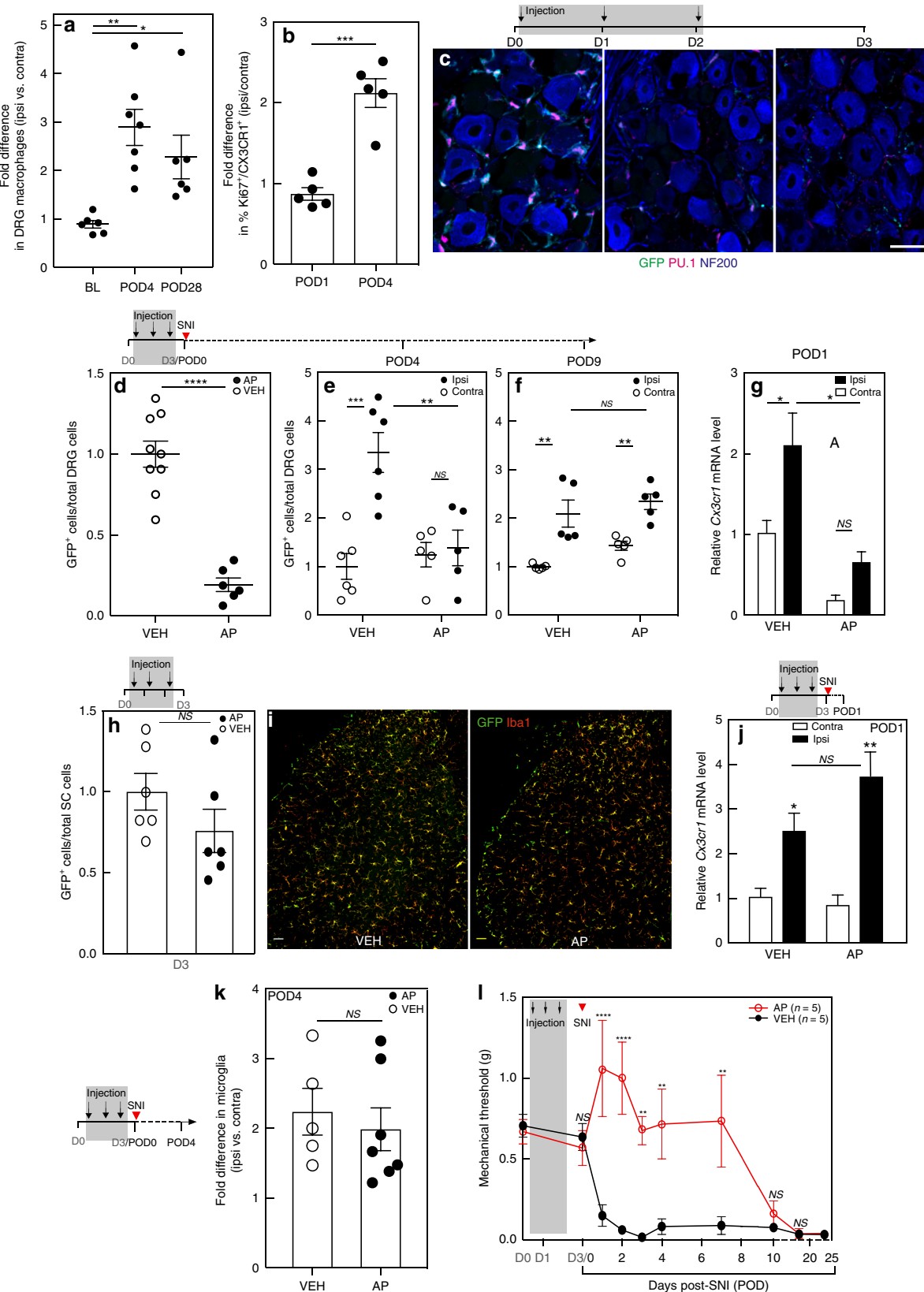

proposed analysis, we considerably tapered the dose and determined that 3 daily 1.0 mg kg$^{-1}$ injections did not affect baseline mechanical (Supplementary Fig. 6B) or thermal (heat) thresholds (Supplementary Fig. 6C, D) in the MAFIA mice. We did not observe deficits in the rotarod test (Supplementary Fig. 6E), although this regimen was still accompanied by weight loss (~15%), compared with vehicle (VEH)-treated mice (Supplementary Fig. 6F).

Using the 3-day protocol, we next used different approaches to examine the fate of resident macrophages in the DRG and of blood monocytes. We monitored DRG macrophages by

**Fig. 1 Systemic AP treatment depletes macrophages in the DRG of MAFIA mice and delays mechanical allodynia. a** FACS analysis of CX3CR1[+] macrophage expansion in the L4/L5 DRG of wild-type mice after SNI ($n = 6$–7 per group). BL, baseline. POD, post-operative day. **b** FACS analysis of Ki67 expression in CX3CR1[+] macrophages in the L4/L5 DRG of wild-type mice after nerve injury ($n = 5$ per group). **c** Representative images illustrating AP-induced depletion of GFP[+] (green)/PU.1[+](red) macrophages in the L4/L5 DRG of MAFIA mice injected with systemic AP (1.0 mg kg[−1]) for 3 days. NF200 (blue) marks myelinated neurons; scale bar: 50 μm. **d** FACS analysis of GFP[+] DRG macrophages 1 day after the 3rd AP injection ($n^{VEH} = 9$; $n^{AP} = 6$). **e, f** SNI 1 day after AP or VEH treatment followed by FACS analysis of DRG macrophages on POD4 (**e**) and POD9 (**f**) ($n = 5$–6 per group). **g** qPCR analysis of *Cx3cr1* gene expression in the L4/L5 DRG of mice pretreated with AP or VEH on POD1 ($n = 3$ per group). **h** FACS analysis of GFP[+] spinal cord microglia after a 3-day AP or VEH ($n = 6$ per group). **i** GFP[+] (green) and Iba1[+] (red) immunoreactive spinal cord microglia after a 3-day AP or VEH. Scale bar: 50 μm. **j** qPCR analysis of *Cx3cr1* gene expression on POD1 in the lumbar spinal cord in mice pretreated with AP or VEH ($n = 3$ per group). **k** FACS analysis of GFP[+] microglia in the lumbar spinal cord of mice pretreated with AP or VEH ($n^{VEH} = 5$; $n^{AP} = 7$) on POD4. **l** Effect on mechanical thresholds of systemic AP or VEH followed by SNI ($n = 5$ per group). Gray shading indicates injection days. Data presented as mean ± SEM. One-way ANOVA with Tukey's correction in **a**, Student's *t*-test in **b, d, h**, and **k**, and two-way ANOVA with Sidak's correction in **e–g, j**, and **l**. *$P < 0.05$, **$P < 0.01$, ***$P < 0.001$, ****$P < 0.0001$, NS nonsignificant compared with control. Source data are available as a Source Data file.

immunostaining for GFP and PU.1, a transcription factor expressed in myelomonocytic cells[26] and in their immature precursors[27]. By the third injection, we recorded a significant loss of GFP[+]/PU.1[+] cells (Fig. 1c) as well as GFP[+]/Iba1[+] cells (Supplementary Fig. 7A, B). In a separate group of mice, 1 day after the 3-day AP regime (D3), FACS analysis demonstrated a significant, ~80% depletion of resident (GFP[+]) macrophage in the L4 and L5 DRG, compared with VEH mice (Fig. 1d). At the same time point, FACS analysis of blood demonstrated an almost 90% depletion of mature (GFP[hi] or CSF1R[hi]) circulating monocytes (GFP[hi]CD11b[+] or GFP[+]CSF1R[hi]), with no change in the VEH-treated group (Supplementary Fig. 7C). Importantly, at 4 days after the last AP injection (D6 in Supplementary Fig. 7D), the percentage of blood monocytes was largely restored, despite a persistent weight loss (Supplementary Fig. 6F). Taken together, these data demonstrate that resident macrophages in the DRG, as well as peripheral monocytic cells, can be significantly, albeit transiently, depleted by AP treatment in MAFIA mice, importantly without altering baseline nociceptive thresholds.

We next examined macrophages in the context of nerve injury. To monitor the early response of macrophages, we followed the expression of *Cx3cr1* in the DRG by quantitative PCR (qPCR) 1 day after the nerve injury (POD1). We recorded a 2.1 ± 0.4-fold increase of *Cx3cr1* expression in the L4/L5 DRG ipsilateral to the nerve injury in VEH-treated mice (Fig. 1g) compared with the uninjured contralateral DRG. On the other hand, AP treatment, prior to the nerve injury, not only reduced baseline *Cx3cr1* expression in the uninjured, contralateral DRG but also prevented the injury-induced increase of *Cx3cr1* expression in the ipsilateral DRG (Fig. 1g). By POD4 (5th day after the last AP treatment), macrophage number recovered to the level recorded in uninjured, VEH-treated mice (Fig. 1e). By POD9, the ipsilateral DRG expansion of macrophages reappeared and did not differ from that recorded in VEH-treated SNI mice (Fig. 1f).

**AP treatment does not alter spinal cord microglia number.** Although previous studies concluded that AP does not cross the BBB[22], we considered it essential to rule out a direct CNS action of the AP treatment. Compared with uninjured, VEH-treated mice, we found no difference in the number of spinal cord microglia by FACS (Fig. 1h) in the lumbar cord of AP-treated mice. Immunostaining for microglia also did not differ (Fig. 1i). These results are consistent with the AP exerting a selective peripheral action. We also addressed the possibility that a compromised BBB following nerve injury could result in AP leakage into the cord[28,29]. Arguing against this possibility, both the nerve injury-induced upregulation of *Cx3cr1* gene expression, evidence of microglia activation, 1 day after SNI (POD1; Fig. 1j) and microgliosis examined 4 days after SNI (POD4; Fig. 1k), did not differ between VEH- and AP-treated mice. Glial fibrillary acidic protein (GFAP) immunostaining of astrocytes also did not

differ (Supplementary Fig. 8). Taken together, these data confirm that neither spinal microglia nor astrocytes are affected by the AP treatment, whether or not there was a nerve injury.

**DRG macrophages are required for initiation of nerve injury-induced mechanical allodynia.** To investigate the contribution of DRG macrophages to nerve injury-induced mechanical hypersensitivity, we treated separate groups of mice with three daily injections of AP (1.0 mg kg[−1]) followed by SNI. In contrast to VEH-treated mice, which developed mechanical hypersensitivity within 24 h of the SNI, the 3-day AP treatment regimen significantly delayed development of the hypersensitivity, for at least 7 days after the nerve injury (Fig. 1l). Importantly, mice in which macrophage expansion was restored exhibited mechanical allodynia. Continuing the AP treatment into the day of nerve injury[6] did not enhance the anti-allodynic effect (Supplementary Fig. 9). These results indicate that the anti-allodynic effect of AP treatment correlates with the prevention of DRG macrophage expansion. Finally, as the AP-induced weight loss persisted in these mice (Supplementary Fig. 6F), we conclude that the AP effect on mechanical hypersensitivity was not secondary to the weight loss. Moreover, because thermal pain thresholds did not differ in the AP-treated mice (Supplementary Fig. 6C, D), we conclude that systemic side effects were not major contributors to the anti-allodynic effect produced by macrophage depletion.

In a separate experiment we transplanted bone marrow (BM) progenitor cells (GFP[+]) isolated from MAFIA mice into irradiated WT mice and confirmed the lack of effect of AP on spinal cord microglia. Specifically, 7 days after the SNI, in contrast to a complete replacement of peripheral host monocytic cells with GFP[+] donor cells (Supplementary Fig. 10A, B), we detected very few GFP[+] cells in the spinal cord (Supplementary Fig. 10C, D). We conclude that the majority of spinal cord microglia (PU1[+]) derived from the host (GFP[−]), even after injury (Supplementary Fig. 10C, D). In a separate group of transplanted mice, we performed SNI after the 3-day AP or VEH treatment. Consistent with our findings in non-transplanted mice, we found that AP-mediated macrophage depletion in the DRG of the transplanted animals (Supplementary Fig. 10E) significantly delayed the development of nerve injury-induced mechanical hypersensitivity (Supplementary Fig. 10F).

**Initiation of the neuropathic pain phenotype requires macrophage expansion in the DRG and is independent of infiltrated macrophages at the nerve injury site.** Next, we examined the relative contribution of macrophage expansion in the DRG and at the nerve injury site. As AP-mediated systemic macrophage depletion cannot distinguish their respective contribution, we designed the following experiment to deplete selectively macrophages at the injury site, leaving DRG macrophages intact. At

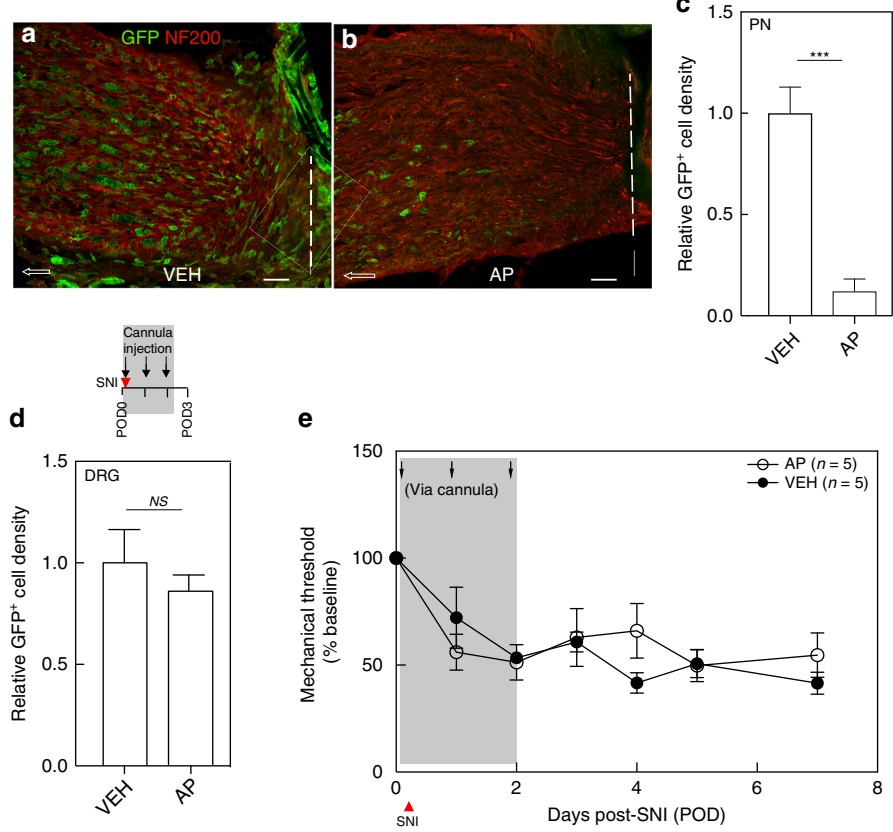

**Fig. 2 Macrophages in the DRG, but not at the nerve injury site, are required for mechanical allodynia initiation. a–e** Effect of SNI followed by cannula injection with VEH or AP on immunostaining of GFP⁺ macrophages at the peripheral nerve (PN) ligature site (**a–c**) and DRG (**d**) (n = 3 per group) and on mechanical thresholds (**e**). Dashed lines in **a** and **b** denote ligature site; arrows point proximally. Scale bar: 50 μm. Gray shading indicates injection days. Data presented as mean ± SEM. Student's *t*-test in **c** and **d**, and two-way repeated-measures ANOVA with Sidak's correction in **e**. ***$P < 0.001$, NS nonsignificant compared with control. Source data are available as a Source Data file.

the time of the nerve injury, we implanted a cannula and sutured it to muscles that overlay the injury site. Our intent was to deliver a low AP dose that only targeted the macrophages around the injury site, with limited to no systemic action. After serial dose titrations, we identified a systemic dose (0.8 μg in 20 μl) that, when administered daily for 3 days, did not influence nerve injury-induced mechanical allodynia (Supplementary Fig. 11A) or cause weight loss (Supplementary Fig. 11B). We administered the low-dose AP via the cannula, prior to performing the nerve injury, and repeated the targeted injection for 2 more days. By quantifying GFP⁺ cell density 1 day after the last injection, compared with VEH-treated mice (Fig. 2a–d), we found an almost 90% reduction of macrophages (GFP⁺) at the nerve injury site in the AP-treated mice (Fig. 2b, c). Notably, this treatment did not prevent macrophage expansion in the injured DRG (Fig. 2d). Most importantly, the targeted AP treatment had no impact on nerve injury-induced mechanical hypersensitivity development (Fig. 2e). Our findings differ from earlier reports that concluded that macrophage expansion at the nerve injury site is the major contributor to neuropathic pain[10,11]. Rather, we conclude that macrophage expansion in the DRG, but not at the site of injury, is required for initiation of the nerve injury-induced mechanical hypersensitivity.

**Macrophages in the DRG contribute to maintenance of nerve injury-induced mechanical allodynia.** SNI-induced mechanical allodynia is typically long lasting and this behavioral phenotype is associated with prolonged microglial activation in the ipsilateral dorsal horn. Figure 3a illustrates, e.g., that microglial activation

persists for at least 4 weeks after peripheral nerve injury. At this time point, we also recorded a persistent increase in the number of macrophages in the ipsilateral DRG (Figs. 1a and 3b) and at the nerve injury site (Fig. 3c).

To study the peripheral macrophage contribution to maintenance of the mechanical hypersensitivity, we repeated the 3-day systemic AP (1.0 mg kg⁻¹) regimen, 28 days after the nerve injury (POD28). In these post-injury studies, by quantifying GFP⁺/PU.1⁺ macrophage density (number of cells per unit area), compared with VEH-treated mice, we recorded an almost 50% decrease of DRG macrophages (Fig. 4a). This decrease was confirmed by qPCR for *Cx3cr1* (Fig. 4b). In a separate group of comparably treated mice, we recorded an almost 80%, albeit transient, reversal of the mechanical allodynia (Fig. 4c). The hypersensitivity reappeared within 6 days of the last AP injection. Finally, we asked whether macrophages at the injury site contribute to maintenance of the persistent mechanical hypersensitivity. We implanted the cannula 4 weeks after the SNI and administered 0.8 μg of AP daily for 3 consecutive days. Figure 4d shows that selective depletion of macrophages at the nerve injury site had no impact on the persistent mechanical hypersensitivity. We conclude that macrophages in the DRG, but not at the peripheral nerve injury site, are also critical contributors to maintenance of the peripheral nerve injury-induced hypersensitivity.

**Nerve injury triggers a reciprocal interaction between DRG macrophages and sensory neurons.** Recently, our laboratory demonstrated that axotomized DRG sensory neurons de novo

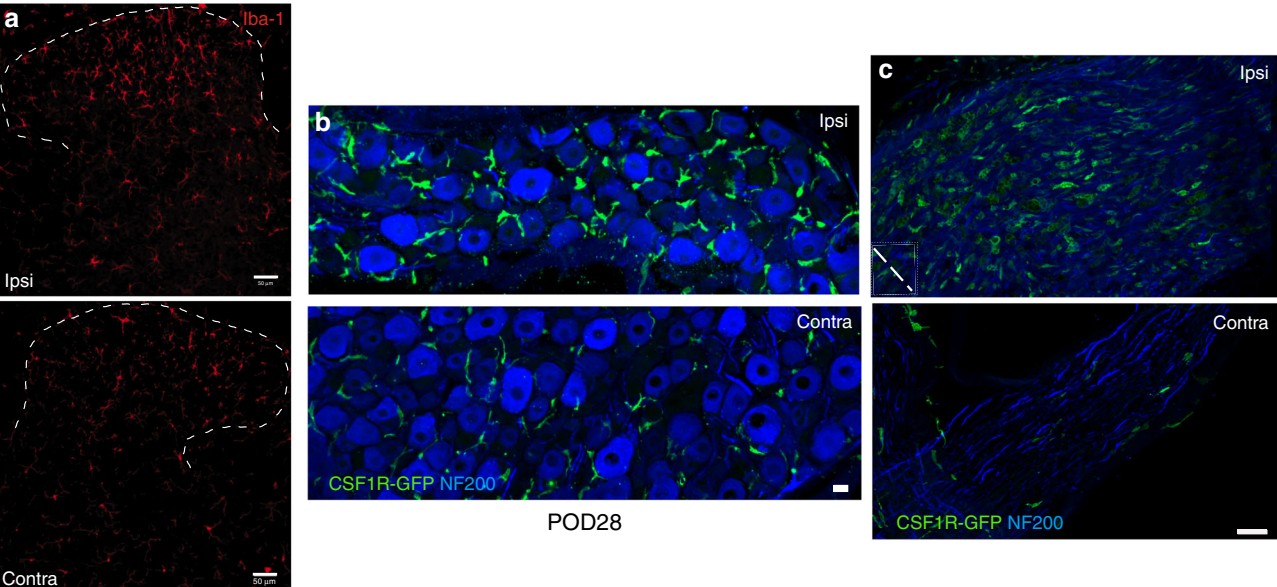

**Fig. 3 Microglia activation and peripheral macrophage infiltration persist at 28 days post SNI. a–c** Immunostaining of Iba1[+] microglia in the dorsal horn (**a**), CSF1R-GFP[+] macrophages (green) in the DRG (**b**), and at the nerve injury site (**c**), 28 days after SNI (POD28). NF200 (blue) labels neuronal cell bodies (**b**) and peripheral nerve myelinated axons (**c**). Dashed line in **c** denotes nerve ligature site. Scale bar: 50 μm in **a**; 15 μm in **b**, **c**.

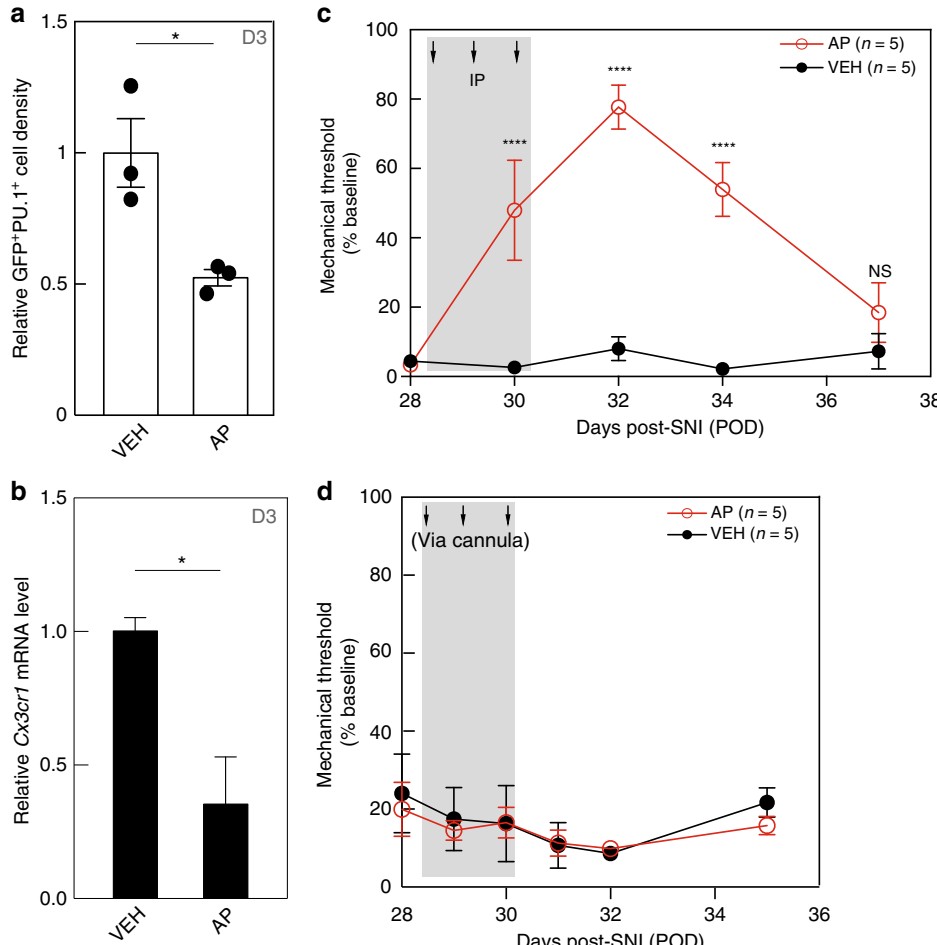

**Fig. 4 Macrophages in the DRG, but not at the peripheral nerve injury site, are required for mechanical allodynia maintenance.** Four weeks after SNI (POD28), MAFIA mice received either systemic (**a–c**) or cannula (**d**) injections of AP or VEH. **a, b** GFP[+]/PU.1[+] macrophage density (**a**) and qPCR of *Cx3cr1* gene expression (**b**) in the ipsilateral DRG 1 day after the 3rd AP injection (D3, $n = 3$ per group). **c, d** Mechanical thresholds after 3-day systemic (**c**) or cannula (**d**) injections ($n = 5$ per group). Gray shading indicates injection days. Data presented as mean ± SEM. Student's *t*-test in **a** and **b**, two-way repeated-measures ANOVA with Sidak's correction in **c** and **d**. *$P < 0.05$, ****$P < 0.0001$, NS nonsignificant compared with control. Source data are available as a Source Data file.

express *Csf1* after peripheral nerve injury[30]. The CSF1, in turn, is transported to the spinal cord dorsal horn, where it activates microglia through the CSF1R, induces microglial proliferation, and promotes the neuropathic pain phenotype. Importantly, in AP-treated mice, the depletion of macrophages did not prevent the injury-induced expression of CSF1 in axotomized sensory neurons (Supplementary Fig. 12A, B). As DRG macrophages also express CSF1R, we next asked whether injury-induced CSF1 contributes to the nerve injury-induced macrophage expansion in the DRG. We performed the analysis in *Adv-Cre; Csf1*$^{fl/fl}$ mice, in which *Csf1* gene expression is depleted selectively from sensory neurons, preventing nerve injury-induced CSF1 expression and the development of mechanical hypersensitivity. As expected, FACS analysis of lumbar dorsal horn microglia 4 days after nerve injury (POD4) showed that the ipsilateral injury-induced microglial activation and proliferation were largely abrogated in the *Adv-Cre; Csf1*$^{fl/fl}$ mice, compared with WT mice (Fig. 5a). Interestingly, conditional deletion of *Csf1* also prevented the injury-induced expansion of macrophages in the ipsilateral, axotomized DRG (Fig. 5b). Finally, the expansion was not compromised in animals globally lacking CCL2, a potent CCR2 ligand (Fig. 5b). We conclude that nerve injury influences DRG macrophage expansion via an interaction with axotomized, CSF1-expressing sensory neurons, and is CCL2 independent.

As we previously reported, CSF1 is also constitutively expressed in satellite cells[30] (Supplementary Fig. 12A, B). As satellite cells in the DRG reportedly contribute to injury-induced neuropathic pain development[8], we also examined the impact of macrophage depletion on satellite cells in the axotomized DRG. Supplementary Fig. 12C, D illustrates that Connexin-43 immunoreactivity, a satellite cell marker[31], was not noticeably altered after AP-mediated macrophage depletion.

We next asked whether the nerve injury-induced DRG macrophage expansion/activation, in turn, influences sensory neurons. Specifically, we examined the impact of depleting DRG macrophages on several well-established markers of genes induced or upregulated in sensory neurons, namely ATF3[32], BDNF[33,34], galanin[35], and neuropeptide Y (NPY)[35]. The analysis was performed in systemic AP-treated mice, 1 day after SNI (POD1). Interestingly, the AP-induced macrophage depletion did not alter the increased expression of *Atf3*, *galanin*, or *Npy* (Supplementary Fig. 13A-C). On the other hand, qPCR analysis showed that the fourfold increase in *Bdnf* mRNA in the axotomized L4 and L5 DRG observed in VEH-treated mice was completely prevented by AP treatment (Fig. 5c). As qPCR could not determine whether the increased *Bdnf* derived only from sensory neurons or also from surrounding non-neuronal cells, we used in situ hybridization (ISH) to examine the DRG. Figure 5d, e shows that upregulation of *Bdnf* mRNA is readily detected in sensory neurons one day after nerve injury (POD1), but we found no expression in surrounding non-neuronal cells, in line with our previous report[36]. Consistent with the qPCR analysis, quantification of ISH intensity in VEH-treated mice demonstrated a threefold induction of *Bdnf* in the ipsilateral compared with the contralateral DRG (Fig. 5d). Also, ISH demonstrated that AP-mediated macrophage depletion significantly attenuated the post-injury *Bdnf* upregulation (Fig. 5e). We conclude that macrophages, which do not express *Bdnf*[36], are nevertheless a contributor to and indeed are required for the nerve injury-induced upregulation of *Bdnf* in axotomized sensory neurons.

To determine how DRG macrophage activation might influence sensory neurons, we next focused on key proinflammatory cytokines, interleukin-1β (IL-1β) and tumor necrosis factor-α (TNFα), which are reportedly expressed in both neuronal and non-neuronal cells of the DRG and have been previously implicated in neuropathic pain[1,37–39]. A qPCR analysis revealed that in VEH-treated mice, on POD1, there is an almost tenfold upregulation of *Il1β* gene expression in the ipsilateral L4/L5 DRG, compared with the contralateral DRG (Fig. 5f). The 3-day AP regimen, performed prior to the SNI, not only abolished the upregulation of *Il1β* in the ipsilateral DRG, but also reduced baseline expression in the contralateral, uninjured DRG. We also used ISH to identify the cells in which there was upregulation of *Il1β* 7 days after nerve injury (POD7; Fig. 5g, h). Although previous immunohistochemical studies reported that injury-induced IL-1β expression occurred in both satellite cells[8] and sensory neurons[40], to our surprise, we only detected *Il1β* message in *Itgam*$^+$ (i.e., CD11b) macrophages (Fig. 5i–l), in both control and nerve-injured animals.

Somewhat unexpectedly, we found no significant *Tnfα* induction in the DRG after nerve injury and macrophage depletion did not alter baseline *Tnfα* expression (Supplementary Fig. 13D). We also examined two putative anti-inflammatory cytokines, namely IL-10[41] and transforming growth factor β (TGFβ)[42,43], exogenous administration of which inhibits nerve injury-induced neuropathic pain behaviors. Based on those reports, we hypothesized that *Tgfβ* levels might decrease in the setting of nerve injury. However, Supplementary Fig. 13E shows that *Tgfβ* gene expression did not change after nerve injury or after macrophage depletion on POD1. *Il10* levels were too low to be detected in the DRG, either before or after injury. These findings suggest that the upregulation of *Il1β* after nerve injury derives predominantly from DRG macrophages and that the anti-allodynic effect of macrophage depletion results, in part, from a reduction of injury-induced *Il1β*.

**The DRG macrophage expansion, but not the contribution to neuropathic pain initiation and maintenance, is sexually dimorphic.** In contrast to the evidence for a contribution of spinal microglia to neuropathic pain in male mice, Sorge et al.[44] reported that ablating microglia in female mice does not prevent the development of mechanical hypersensitivity after peripheral nerve injury. Here we asked whether the contribution of DRG macrophages to nerve injury-induced mechanical hypersensitivity is also sexually dimorphic. FACS analysis of DRG in female WT mice 4 days after nerve injury (POD4) showed a significant increase (1.69 ± 0.21-fold) in macrophages in the DRG ipsilateral to the injury, compared with the contralateral, uninjured side (Fig. 6a). This increase, although significant, was nevertheless not as robust as in the male mice (Fig. 1a). On the other hand, although conditional deletion of *Csf1* prevented the macrophage expansion in male mice, neither conditional sensory neuron deletion of *Csf1* nor global deletion of CCL2 in female mice prevented the macrophage expansion (Fig. 6a). Clearly, additional axotomized sensory neuron-derived factors must influence macrophage expansion in female mice.

We next directly tested the contribution of the macrophages to SNI-induced mechanical hypersensitivity in female mice. As in the male mice, macrophage depletion by three daily AP injections (Supplementary Fig. 14A) delayed development of the mechanical hypersensitivity (Fig. 6b). In addition, as in the male mice, spinal microglia in the female mice were not affected (Supplementary Fig. 14B). Furthermore, macrophage depletion by AP administration 4 weeks after the nerve injury, when there was demonstrable mechanical allodynia, also produced a transient, albeit smaller (60%) reversal of the allodynia in the female mice (Fig. 6c). We also examined both male and female animals lacking CCL2 and found that SNI-induced mechanical hypersensitivity was comparable in male and female mice (Fig. 6d). Taken together, we conclude that DRG macrophages contribute to the initiation and maintenance of nerve injury-induced mechanical hypersensitivity in both male and female mice, but that there is sexual dimorphism in the contribution of CSF1 to nerve injury-induced expansion of DRG macrophages.

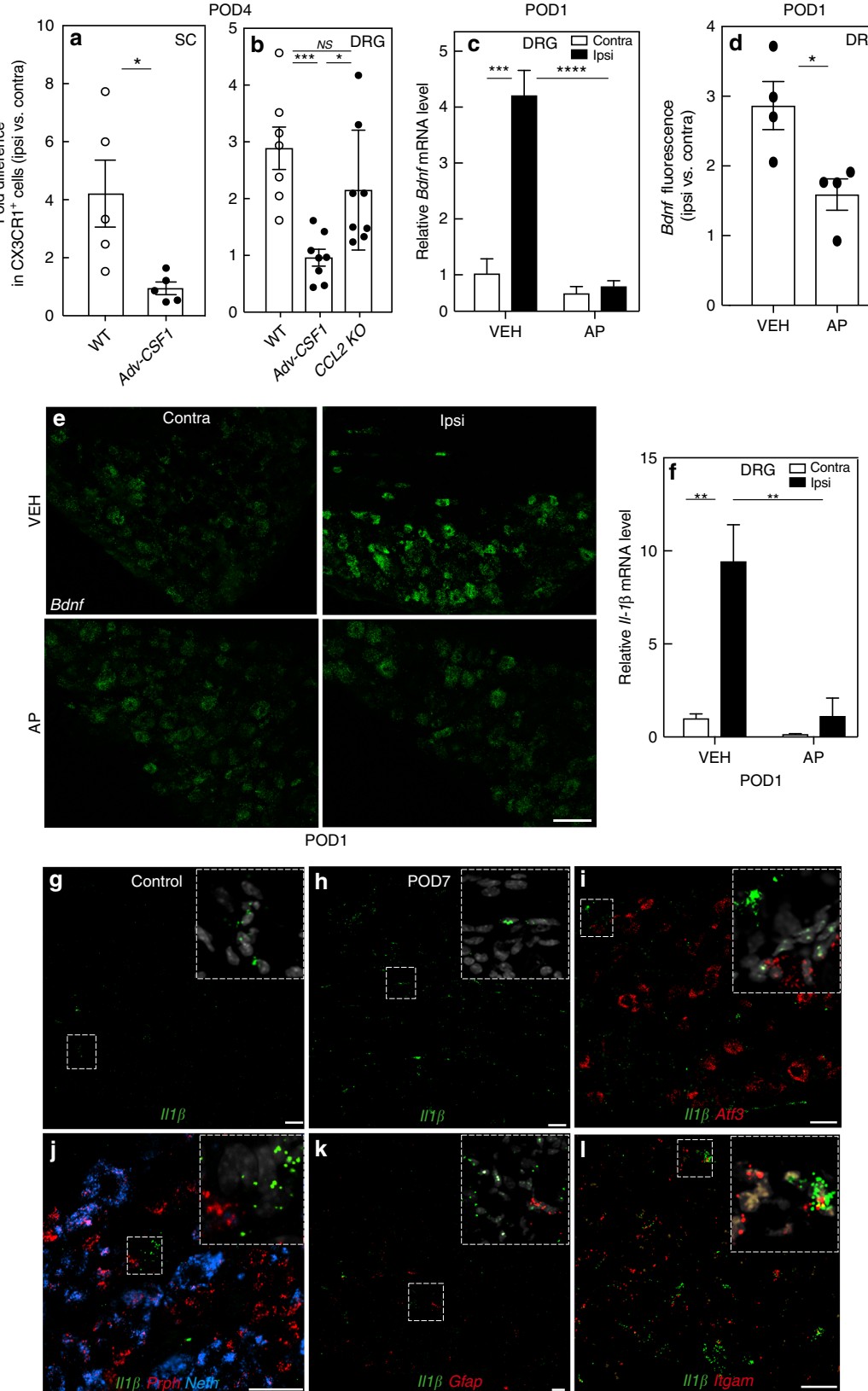

## Discussion

Although several groups have implicated peripheral macrophages in nerve injury-induced pain initiation[5,21], there is little consensus as to which macrophage population, in the DRG or at the nerve injury site, is most relevant. As selective depletion of DRG macrophages is difficult, here we instead targeted macrophages at the injury site, using an implanted cannula that delivered a low-dose of AP, one that killed macrophages at the injury site, but had no systemic effect, thus sparing the DRG. Based on the profound effect of systemic AP and the lack of effect of the sciatic nerve-

**Fig. 5 Nerve injury triggers reciprocal molecular interactions between DRG macrophages and sensory neurons. a** FACS analysis of CX3CR1+ microglia in the lumbar spinal cord of WT and Adv-*Csf1* mice 4 days after SNI (POD4, $n = 5$ per group). **b** FACS analysis of CX3CR1+ macrophages in the L4/L5 DRG of WT ($n = 7$), Adv-*Csf1* ($n = 8$), and CCL2 knockout (KO) mice ($n = 8$) on POD4. **c–e** qPCR (**c**, $n = 3$ per group), or in situ hybridization (ISH) (**d**, **e**, $n = 4$ per group) analysis of *Bdnf* expression in the DRG of mice pretreated for 3 days with systemic AP or VEH on POD1. Scale bar: 15 μm in **e**. **f** qPCR for *Il1β* on POD1 in the DRG of mice pretreated with AP or VEH. **g–l** DRG expression of *Il1β* mRNA (green) in control (**g**) or 7 days after nerve injury (POD7, **h-l**). **i–l** Coexpression of *Il1β* with neuronal markers *Atf3* (**i**), *Prph* (red, **j**), *Nefh* (blue, **j**); a satellite cell marker, *Gfap* (red, **k**); and a macrophage marker, *Itgam* (red, **l**). Scale bars: 50 μm. Insets illustrate high magnification of labeled cells. Data presented as mean ± SEM. Student's *t*-test in **a** and **d**, one-way ANOVA with Tukey's correction in **b**, and two-way ANOVA with Sidak's correction in **c** and **f**. *$P < 0.05$, **$P < 0.01$, ***$P < 0.001$, ****$P < 0.0001$, NS nonsignificant compared with control. Source data are available as a Source Data file.

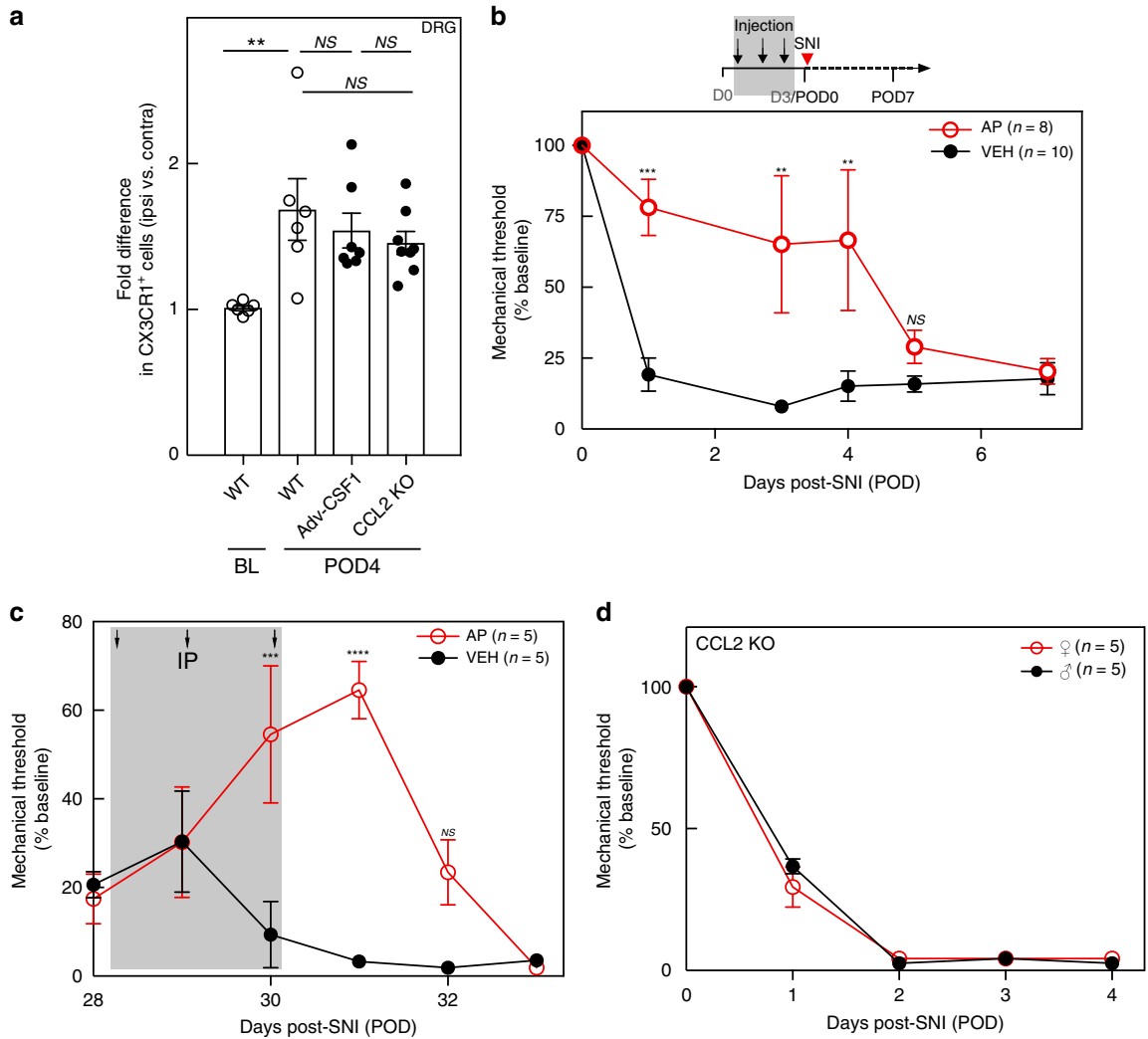

**Fig. 6 The DRG macrophage contribution to neuropathic pain initiation and maintenance is not sexually dimorphic. a** FACS analysis of ipsilateral CX3CR1+ macrophage expansion in the L4/L5 DRG of WT ($n = 6$), Adv-*Csf1* ($n = 7$), and CCL2 KO ($n = 8$) female mice on POD4. **b** Effect on mechanical thresholds of systemic AP ($n = 8$) or VEH ($n = 10$) followed by SNI in female MAFIA mice. **c** Effect on mechanical thresholds of systemic AP or VEH ($n = 5$ per group) 4 weeks after SNI in female MAFIA mice. **d** Effect of CCL2 deletion on mechanical thresholds in male and female mice ($n = 5$ per group). Gray shading indicates injection days. Data presented as mean ± SEM. One-way ANOVA with Tukey's correction in **a** and two-way repeated-measures ANOVA with Sidak's correction in **b–d**. *$P < 0.05$, **$P < 0.01$, ***$P < 0.001$, ****$P < 0.0001$, NS nonsignificant compared with control. Source data are available as a Source Data file.

targeted injection, we conclude that DRG macrophages, but not those at the nerve injury site, are critical contributors to both the initiation and maintenance of the mechanical hypersensitivity hallmark of neuropathic pain in mice. Although we cannot rule out a contribution of a systemic treatment-induced depletion of dendritic cells in the skin, which are also killed by AP[22], we favor the view that DRG macrophages are the major contributor.

Not only did we demonstrate that macrophage depletion prevented the nerve injury-induced mechanical allodynia, but we also found that the mechanical allodynia reappeared after restoration of the injury-induced ipsilateral expansion of macrophages. A critical question is what drives the DRG macrophage expansion in the first place. We previously reported that activation of dorsal horn microglia requires de novo expressed CSF1 in

axotomized DRG sensory neurons, and that its selective deletion from sensory neurons prevents the neuropathic pain phenotype[30]. Here we found that conditional deletion of CSF1 from sensory neurons of male, but not female, mice also prevents the injury-induced expansion of DRG macrophages, which suggests that the effect of CSF1 deletion on the neuropathic pain phenotype in the male mouse reflects an integrative action on both DRG macrophages and dorsal horn microglia.

We also addressed the possible source of the injury-induced macrophage expansion in the DRG. Although CCR2 expression has been associated with infiltrating macrophages[23], we found many CCR2$^+$ resident macrophages in the DRG of uninjured mice. We conclude that CCR2 expression in DRG macrophages is not a suitable marker that can distinguish infiltrating from resident macrophages. Furthermore, using Ki67 to document proliferating cells, we found that the percentage of Ki67$^+$CX3CR1$^+$ macrophages in the axotomized DRG more than doubled. Based on these results, although we cannot rule out a contribution from infiltration, we favor the view that macrophage expansion originates predominantly from resident DRG macrophages that proliferate after injury.

Of particular interest is our finding that activated DRG macrophages reciprocally influence sensory neurons after injury. Specifically, depletion of DRG macrophages prevented the upregulation of *Bdnf* in the DRG that occurred within 24 h of SNI. Importantly, despite reports of BDNF synthesis by microglia[45] and presumably macrophages, consistent with other findings[36,46], our ISH analysis only detected *Bdnf* mRNA in sensory neurons. On the other hand, as conditional deletion of BDNF from sensory neurons has minimal to no effect on the mechanical allodynia induced by nerve injury[34,36], to what extent the macrophage contribution to *Bdnf* upregulation in sensory neurons influences nociceptive processing is not clear.

The question remains, therefore: how does the DRG macrophage influence the sensory neuron contribution to nerve injury-induced mechanical allodynia? Triggered by peripheral nerve injury, immune cells, including macrophages, release abundant proinflammatory mediators that directly or indirectly induce pain hypersensitivity[1,39,47]. For example, upregulation of *Il1β* in both DRG and spinal cord after nerve injury is implicated in the central sensitization process[1,48–50]. Furthermore, injection of either an IL-1-receptor antagonist[51] or a neutralizing antibody to IL-1β[52] can reduce neuropathic pain behavior in mice. Here we found that both the baseline and the rapidly induced *Il1β* mRNA expression in the DRG 1 day after SNI are significantly prevented by macrophage depletion. Interestingly, although previous immunohistochemical reports demonstrated that both satellite cells[8] and sensory neurons[40] are the main source of injury-induced IL-1β expression in the DRG, we could not confirm those findings using ISH. Instead, we detected *Il1β* induction in the CD11b$^+$ macrophages, consistent with our reverse-transcriptase PCR findings after macrophage depletion. Our results suggest that DRG macrophages are the main source of the *Il1β* increase in the DRG after nerve injury, and that IL-1β, in turn, contributes to the sensitization of sensory neurons.

Our findings are also of interest in light of the report of Sorge et al.[44] that microglial depletion in male, but not in female mice, reduced nerve injury-induced neuropathic pain. Here we found that depleting DRG macrophages not only prevented the development but also maintenance of mechanical hypersensitivity, in both male and female mice. Although the contribution of the DRG macrophages to the mechanical hypersensitivity appears not to be sexually dimorphic, there were some important differences noted between male and female mice. First, the magnitude of the injury-induced macrophage expansion in male mice was double that observed in female mice. This difference conceivably

underlies the shorter duration of the AP-mediated anti-allodynic effect in female vs. male mice (4 vs. 7 days). Furthermore, only in male mice did deletion of CSF1 in sensory neurons influence macrophage expansion in the DRG. We conclude that other, as yet unidentified, sensory neuron-derived factors differentiate male and female mice.

In conclusion, we have demonstrated that macrophages in the DRG of both male and female mice contribute to and are required for neuropathic pain initiation and maintenance. We also uncovered a reciprocal cellular interaction between macrophages and sensory neurons in the DRG, an interaction that in concert with the sensory neuron–microglia connection, contributes to the neuropathic pain phenotype. To what extent these processes are independent and whether they can be targeted together as a novel pain therapeutic remains to be determined.

## Methods

**Animals**. Adult mice (4–12 weeks old) were used in all experiments. WT C57BL/6 mice, MAFIA (CSF1R-EGFP-NGFR/FKBP1A/TNFRSF6) transgenic mice[22] (Stock #005070), CCL2 knockout mice[53] (Stock #004434), and CCR2-RFP$^{+/+}$ knock-in mice[54] (Stock #017586) were obtained from the Jackson Laboratory. We also studied CX3CR1$^{CreER-EYFP}$ mice[18] originally generated by Wen-Biao Gan at New York University. We crossed homozygous MAFIA mice with homozygous CCR2-RFP$^{+/+}$ mice to generate CSF1R-GFP$^{+/-}$CCR2-RFP$^{+/-}$ mice. All animal experiments were approved by the Institutional Animal Care and Use Committee at University California San Francisco and were conducted in accordance with the NIH Guide for the Care and Use of Laboratory animals.

**AP administration**. The AP (Clontech, #635058) was diluted in a distilled water solution consisting of 4% ethanol, 10% PEG-400, and 1.7% Tween 20[22], and injected intraperitoneally.

**Immunohistochemistry**. Mice were anesthetized with 2.5% Avertin and perfused transcardially with 1× phosphate-buffered saline (PBS) followed by 4% formaldehyde. Dissected tissues were first post-fixed in the same fixative for 3 h, then preserved overnight in 30% sucrose in PBS before cryostat sectioning. The following antibodies were used to immunostain DRG, spinal cord, and peripheral nerve sections: chicken anti-GFP (1:2000, Abcam #ab13970), rabbit anti-CSF1R (1:15,000, Millipore #06–174), rabbit anti-Iba1 (1:2000, Wako #019–19741), rabbit anti-PU.1 (1:500, Cell Signaling #2266), mouse anti-NF200 (1:1000, Sigma #N5389), rabbit anti-dsRed (1:500; Clontech #632393), rabbit anti-Connexin-43 (1:2000, Sigma #C6219), rabbit anti-GFAP (1:20,000, DAKO #Z0334), goat anti-CSF1 (1:500, R&D #AF416), and fluorophore-coupled secondary antibodies (1:1000, Alexa Fluor 488, 555, 594, 647, ThermoFisher Scientific). Images were captured with a Carl Zeiss LSM 700 microscope and processed with Fiji/ImageJ (NIH).

**In situ hybridization**. We used the RNAscope Fluorescent Multiplex Reagent Kit (Advanced Cell Diagnostics) according to the manufacturer's instructions. Briefly, freshly dissected tissue was quickly frozen on dry ice, cryostat sectioned at 12 μm, and mounted on slides. The mounted sections were fixed in prechilled 10% neutral-buffered formalin for 15 min at 4 °C. After a series of dehydration steps in gradient ethanol solutions, the sections were pretreated with Protease K for 30 min at room temperature, and then incubated with RNA probe for 2 h at 40 °C in an HybEZ$^{TM}$ oven. Repeated washing and amplifier hybridizations were performed according to the manufacture's protocol. Finally, sections were costained with DAPI (4′,6-diamidino-2-phenylindole) before mounting. The following probes were used: *Bdnf* (#424821), *Il1β* (#316891), *Atf3* (#426891), *Prph* (encoding Peripherin protein, #40036), *Nefh* (encoding NF200 protein, #443671), *Itgam* (encoding CD11b protein, #311491), and *Gfap* (#313211).

**Image quantification**. We analyzed every 5th immunostained section through the L4 and L5 DRG using ImageJ[30]. To quantify the density of GFP$^+$PU1$^+$ monocytic cells in the DRG and of microglia cells in the spinal cord, we counted all labeled PU.1$^+$ cells using ImageJ. We also used ImageJ to measure the density of GFP immunoreactivity in DRG macrophages. To quantify macrophage density at the sciatic nerve injury site, we used ImageJ to assess GFP immunoreactivity within an area extending 500 μm proximal to the ligature. To quantify the *Bdnf* ISH message, we used ImageJ to analyze signal intensity in a sensory neuron-rich region of the DRG (five to eight sections from the same ganglia; four mice per group). An individual blind to the treatment groups performed the image analysis.

**Quantitative real-time PCR**. qPCR analyses were carried out with gene specific primers and fluorescent labeled Taqman probes (ThermoFisher Scientific) for *Cx3cr1* (Mm00438354_m1), *Bdnf* (Mm04230607_s1), *Il1β* (Mm00434228_m1),

*Tgfβ* (Mm01178820_m1), *Il10* (Mm01288386_m1), *Tnfα* (Mm00443258_m1), *Atf3* (Mm00476033_m1), *Gal* (Mm00439056_m1), and *Npy* (Mm01410146_m1). Relative expression level was calculated using the $2^{-\Delta\Delta CT}$ method[55]. β-Actin (Mm02619580_g1) was used as the internal control for each sample.

**Isolation of spinal microglia, DRG macrophages, and peripheral blood cells**. Peripheral blood was drawn from a retro-orbital vein in isoflurane anesthetized mice using heparinized micropipettes (ThermoFisher Scientific #22362566). Red blood cells were lysed first before immunostaining. Fresh isolated spinal cord tissue was digested with collagenase, dissociated, and filtered through cell strainers (70 μm). Microglia were further enriched using a myelin removal beads protocol (Miltenyi Biotech #130–096–733). Alternatively, freshly dissected lumbar cord or L4/L5 DRG was first homogenized in cold calcium and magnesium-free Hanks' balanced salt solution. After filtering through a cell strainer (70 μm), the cell homogenate was mixed with Percoll (Sigma) for myelin removal and enrichment[56].

**Flow cytometric assays**. Dissociated cells were briefly fixed in 4% paraformaldehyde for 15 min at room temperature and washed once with FACS staining buffer (5% fetal bovine serum (FBS) in PBS). Cells were then resuspended in 100 μl of saponin buffer (0.5% saponin and 2% FBS in PBS) and immunostained with anti-CD115-PE cy7 (1:1000, eBioscience #25115282), CD11b-APC-cy7 (1:2000, Biolegend #101226), PU.1 (1:1000, Cell Signaling #2266), chicken anti-GFP antibody (1:1000, Abcam #ab13970), anti-CX3CR1-APC (1:2000, Biolegend #149008), or anti-Ki67-PerCP-efluro710 (1:2000, Invitrogen, #66–5698–82) for 60 min at 4 °C[57].

**Surgery and behavioral analyses**. For the SNI model of neuropathic pain[58], under isoflurane anesthesia, we ligated and transected the sural and superficial peroneal branches of the sciatic nerve, leaving the tibial nerve intact. For cannula implantation, after the sciatic nerve branches were exposed, a polyethylene capillary tubing (1.0 mm OD, Sutter Instrument, Novato, CA) was embedded next to the nerve, sutured to overlying muscle and secured between two skin staples. All behavioral experiments were performed as previously reported in a blinded manner during the light cycle[59].

**Bone marrow transplantation**. Donor BM cells from CSF1R-GFP mice (CD45.2+) were collected at 4–5 weeks of age, enriched by immunomagnetic depletion of cells expressing mature hematopoietic lineage antigens defined by a cocktail of monoclonal antibodies: CD5 (Ly-1), CD11b (Mac-1), CD45R (B220), Gr-1, and TER119 (#19856 A, StemCell Technologies, Vancouver, BC, Canada). Cells were then transplanted through a retro-orbital vein into lethally irradiated (1,100 cGy) recipient B6.SJL (CD45.1+) mice (NCI). Four weeks after the transplantation, and prior to further study, engraftment was determined by FACS analysis of peripheral blood.

**Statistical analysis**. Data are expressed as mean ± SEM. Statistical analysis was performed using GraphPad Prism version 7.0 (GraphPad Software). Student *t*-tests were used for single comparisons between two groups. Other data were analyzed using one-way or two-way analysis of variance. *$P < 0.05$, **$P < 0.01$, ***$P < 0.001$, ****$P < 0.0001$, NS nonsignificant.

**Reporting summary**. Further information on research design is available in the Nature Research Reporting Summary linked to this article.

## Data availability
Source data underlying Figs. 1, 2, 4, 5, 6 and Supplementary Figures 2, 5, 6, 7, 9, 10, 11, 13, 14 are a available as a Source Data file. All other data supporting the findings of this study are available from the corresponding author upon reasonable request.

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

## Acknowledgements

The study was supported by the following: Foundation for Anesthesia Education and Research (X.Y.); the UCSF Department of Anesthesia and Perioperative Care (X.Y.); NSR35097306 (A.B.), Wellcome Award: A102645 (A.B.), and NS100801–01 (G.Z.). This study was supported in part by HDFCCC Laboratory for Cell Analysis Shared Resources Facility through a grant from the NIH (P30CA082103).

## Author contributions

X.Y., H.L., K.A.H., M.M., S.Y., J.L. and J.M.B. performed the experiments and collected the data. Z.G. assisted with experimental design. X.Y. and A.I.B. designed the study and analyzed data. X.Y., J.M.B. and A.I.B. wrote the manuscript.

## Competing interests

The authors declare no competing interests.
