## [Peer Review File · Nature Communications]

Reviewers' comments:

Reviewer #1 (Remarks to the Author):

In this study, Yu et al. investigated the contribution of DRG macrophages to peripheral nerve injury-induced neuropathic pain. There are several novel findings in this study: (1) Using an interesting mouse line (MAFIA), the authors were able to selectively deplete peripheral macrophages without impacting microglia. The depletion of macrophages attenuated both initiation and maintenance of SNI-induced mechanical hypersensitivity. (2) Using local AP to kill macrophages at the injury site (but had no systemic effect), the authors found that DRG macrophages but not those at the nerve injury site are critical contributors to neuropathic pain. (3) Conditional deletion of CSF1 from sensory neurons prevents the injury-induced expansion of DRG macrophages. On the other hand, macrophage deletion attenuated SNI-induced BDNF expression in sensory. (4) Given the reported sexual dimorphism in the microglial contribution to pain filed, the authors examined the role of DRG macrophages in both male and female mice. They found that SNI-induced expansion of DRG macrophages in both male and female mice, and depletion of DRG macrophages reduced SNI-induced chronic pain in both male and female mice. Overall, the results are somewhat surprising, but very exciting. The experiments were well designed and nicely performed. The study will shed new light on the role of peripheral macrophages in neuropathic pain and thus may provide potential pain therapeutics. Some concerns need to be addressed:

Major concerns:

1. Since the systemic AP treatment also depletes most of the dendritic cells and other tissue macrophages, the transient pain relief may come from sites other than DRGs, for example, the nerve fiber between DRG and injury site, the skin where the nerve fiber terminals located. Although the authors titrated the AP dose, other side effects cannot be completely ruled out. Indeed, the authors observed the weight loss after systemic AP application (Fig S4F). Therefore, the authors may need to discuss the possible other cell ablation than DRG macrophages as well as the complication of side effects of whole-body macrophages depletion.
2. The results are quite exciting that macrophages deletion without microglia ablation can attenuate SNI-induced pain hypersensitivity for at least a week. Particularly, it is interesting that increased microglial number still persists when pain is attenuated (Fig 2). Three questions: (1) wondering whether SNI-induced spinal microglial proliferation (Gu et al., Cell Rep, 2016) might be altered after DRG macrophage deletions; (2) similarly, how about SNI-induced CSF1 expression in the DRGs (Guan et al., Nat Neurosci, 2016)? (3) and how about the activation of DRG satellite cells or spinal astrocytes after macrophage deletion?
3. It is confusing to put together Fig 3A with the rest of figure, because Fig 3A characterizes behaviors after DRG macrophage deletion while the rest of Fig 3 describes macrophage deletion at injury sites. It might be okay to have Fig 1, 2, and Fig 3A in the same figure.
4. Since macrophages repopulate quickly after ablation (presumably less one week), it is not clear why the pain hypersensitivities came back long after macrophages restored. It is possible that the ablation time duration was not long enough to cover the critical time window for the neuropathic pain development. Wondering whether prolonged AP treatment (to cover SNI treatment day) may completely reverse the chronic pain development as shown by both microglia/macrophages ablation (Peng et al., Nat Neurosci, 2016)? Also, it would be good to add a more time points describing the time course of macrophage depletion and repopulation in DRGs.
5. Along the similar line, the macrophage depletion and repopulation in DRGs should be shown to examine the role of macrophages in the maintenance of neuropathic pain (Fig. 4). The authors showed macrophage deletion can reverse the mechanical hypersensitivity between POD28 to POD37, but did not correlate the restoration of pain with macrophage repopulation.
6. For the source of macrophage expansion in DRGs, the authors showed some conflicting results here. If the authors wanted to claim a major source from infiltration, immunostaining and the cell densities for both resident and infiltrated macrophages should be performed in addition to the FACS data (Fig S7). After SNI, the resident macrophages in DRG seem still dominant, thus the proliferation could contribute more for the cell expansion. Consistent with this notion, the authors

then showed that conditional deletion of CSF1 from sensory neurons abolished SNI-induced expansion of DRG macrophages (Fig. 4B). Unless CSF1 is able to induce CCR2+ macrophage proliferation (which is unlikely as they do not express CSF1R), the results on the source macrophage expansion after SNI are very confusing.

7. The claim on IL1b dependent mechanism is relatively weak (Fig 5F) as DRG macrophages presumably produce a slew of proinflammatory cytokines after SNI. To strengthen the conclusion, a few additional experiments would be helpful: (1) A cytokine array would be good to screen the overall changes in a variety of cytokines/chemokines; (2) In addition to RT-PCR, it would be better to use immunostaining to show the increased expression of IL1b in DRG macrophages after SNI; (3) More importantly, it would be very convincing that IL1b local injection in DRGs can rescue the pain phenotype after DRG macrophage deletion.

Minor concerns:

1. Page 3, 2nd paragraph, "Chlodronate" should be "Clodronate".
2. Page 4, it would be better to introduce that administration of AP in MAFIA mice selectively depletes macrophages but not microglia due to its impermeability to BBB.
3. Fig 5, the authors might want to tone down BDNF results as BDNF is the mechanism for the pain phenotype in macrophage deletion mice. Fig 5E, there is no labeling for BDNF.
4. The mechanical pain threshold in this study (around 0.5g) is much lower than usual, even their own study (e.g., around 1g in Guan et al., 2016). The author should explain the discrepancy.
5. Fig S6C-D needs better descriptions in the figure legends and in the results.

Reviewer #2 (Remarks to the Author):

This paper investigated the role of macrophages in the DRG in neuropathic pain using transgenic mice that enable selective depletion of macrophages. They show that the initiation and maintenance of nerve injury-induced mechanical hypersensitivity are reduced by depleting macrophages in the DRG, but not at the site of injury. Nerve injury increases the number of DRG macrophages whose source could be circulating monocytes. The increase in DRG macrophages is suppressed by conditional knockout of CSF1 in DRG neurons. The authors also show no sexual dimorphism in the expansion of DRG macrophages after nerve injury and in the contribution to mechanical hypersensitivity, although fewer macrophages are induced in females. From these findings, the authors conclude that macrophages in the DRG, but not injured site, contribute to both the initiation and maintenance of neuropathic pain. While this is an interesting paper providing evidence for the role of DRG macrophages, I have serious concerns about key aspects of the paper.

Major points

1. Using a technique enabling a selective depletion of macrophages at the site of injury, the authors clearly show that these cells have no contribution to nerve injury-induced mechanical hypersensitivity. Under this experimental condition, macrophages could be intact in the DRG, leading to one of the main conclusions of this paper that DRG macrophages are critical contributors to neuropathic pain. However, the local treatment of AP at the site of injury might also spare macrophages in the sciatic nerve that have previously been implicated in neuropathic pain. Thus, it remains unclear whether the effect of AP on neuropathic pain is dependent on macrophage depletion in the DRG. Therefore, to clearly show the role of macrophages in the DRG, the authors should examine neuropathic pain initiation and maintenance using mice whose DRG macrophages are selectively depleted by local application of AP into the DRG. This is essential to strengthen the conclusion of this paper.

2. The mechanism for the expansion of macrophages in the axotomized DRG is also unclear. From the data in Figure S7A,B showing an increase in CCR2+CSF1R+ macrophages in the DRG, the

authors conclude that the injury-induced expansion results from infiltrating circulating monocytes. However, it seems that the injury-induced increase in CCR2⁺ macrophages in the DRG was slight, and infiltrating macrophages/monocytes could not be a major population in total DRG macrophages. The authors need to include immunohistochemical data of CCR2⁺CSF1R⁺ macrophages in the DRG before and after nerve injury. Also, the percentage of GFP⁺ cells in the DRG of bone marrow chimeric mice after the injury should be counted. Furthermore, proliferation of macrophages should also be tested using proliferation markers such as BrdU.

3. The authors conclude that systemic AP treatment had no effect on spinal cord microglia. The number of microglia in the spinal cord of AP-treated mice without injury is not changed by AP (although it was slightly decreased), but it is more important to show that the AP treatment does not affect the nerve injury-induced microgliosis because this is a crucial step for activation of microglia and neuropathic pain. Thus, the authors should examine the number of spinal cord microglia and the expression of CX3CR1 mRNA at 4 days after nerve injury in vehicle and AP-treated mice.

4. The mechanism for the cellular interaction between macrophages and sensory neurons in the DRG is still unclear. The authors show upregulation of BDNF and IL-1b, but how do they interact and contribute to macrophage expansion or CSF1 upregulation? Also, is CSF1 upregulation in axotomized DRG neurons changed by macrophage depletion?

Minor points

1. In Figure 6B, DRG macrophage depletion by AP in females should be given.
2. Two identical Figure S4 are included in this paper.
3. Peng et al. have recently demonstrated that depletion of circulating monocytes does not change the injury-induced increase in macrophages in the DRG (Nat Commun 7: 12029, 2016). An explanation for this discrepancy should be included in Discussion.
4. CCR2-knockout mice (CCR2-REF/REF mice), but not CCL2-knockout mice, should be used for examining the nerve injury-induced macrophage infiltration in the DRG.
5. Satellite glia has been proposed to be a source of IL-1b in the DRG after nerve injury (Pain 158: 1666-1677, 2017). The immunohistochemical or in situ hybridization analysis should be performed to determine the IL-1b-expressing cells in the DRG.

Reviewer #3 (Remarks to the Author):

In this article, Yu and colleagues investigate the contribution of macrophage infiltration into the peripheral nervous system to both the initiation and persistence of neuropathic pain, using the spared nerve injury (SNI) model in mice. Using a transgenic mouse line enabling inducible depletion of macrophages without targeting microglia, they demonstrate that macrophages accumulating in the DRG, but not those at the site of nerve injury, contribute to the initiation and maintenance of mechanical hypersensitivity/neuropathic pain. Further, they report that depletion of DRG macrophages reduced this mechanical hypersensitivity in male and female mice. They observe an interesting sexual dimorphism in the requirement of neuronal CSF1 for macrophage expansion, which was only observed in male mice. The origins of the macrophages contributing to the expansion in the DRG are also addressed.

The manuscript addresses several unresolved topics in the field of neuroinflammatory macrophage/microglia crosstalk in chronic pain. They employ existing methods in new ways to arrive at conclusions that will undoubtedly be of interest to others in the community. Statistical analyses appear to be appropriate throughout, and the level of experimental detail seems sufficient to allow researchers to reproduce the work. The manuscript is well-written and the figures well-presented.

1. Microglia express CSF1R, the promoter of which controls the MafIA receptor expression. What

happens to Iba1 expression following AP administration in the DRG, is that reduced also?

2. In Figure S5A, there seems to be a large increase in the CD11b+ GFPlo population following AP administration. Does this represent monocytes with low-level expression of CSF1R, or some other cell type perhaps? This could have a meaningful impact on the cell types infiltrating the PNS following AP administration and nerve injury.

3. Have the authors employed some form of positive control to see if activation of sensory neurons is still able to elicit pain following AP depletion of monocytes/macrophages? Such a control would ensure that neurons are still capable of relaying a nociceptive stimulus after AP. This would lend credence to the idea that apparent prevention/loss of hypersensitivity is due primarily to loss of macrophages in the context of SNI, rather than some off-target effect e.g. lethargy/cachexia.

4. Figure 3. Why are the values in 3A depicted as absolute threshold, yet % of baseline is used for all subsequent assessments of mechanical sensitivity? Furthermore, can the authors explain why the percent reduction from baseline is only 50% or so in Figure 3E, yet it routinely appears to be much greater - 5-10% of baseline in other figures?

5. The authors remark that local delivery of AP to the future site of injury did not prevent subsequent macrophage expansion in the DRG (Figure 3D). The right panel in figure 3D appears to show no significant difference between vehicle and AP treatment. I am guessing that the 'veh' value (normalized to 1) already reflects the macrophage expansion reported in earlier figures, and thus the reported lack of significant difference between the 'veh' and 'AP' values indicates DRG macrophage expansion was unaffected by prior AP?

6. The authors theorize that DRG macrophages are the main source of the observed increase in IL-1 beta, which facilitates sensory neuron sensitization in response to injury. Similar increases have also been reported in peripheral nerve lesions (e.g. Kleinschnitz C et al., J. Neuroimmunol 149 (2004) 77-83), the existence of which should be discussed.

7. This brings me to another point which should be discussed; could the authors speculate on whether there is an intrinsic difference in the function of macrophages that infiltrate the peripheral nerve versus the DRG which may underlie their results, or do they think that inflammatory mediators produced at the site of nerve injury cannot excite sensory afferents to the same extent as those adjacent to somata in the DRG?

8. The authors present evidence that macrophages at the site of nerve injury do not contribute to neuropathic pain, contrary to the study by Shepherd et al., however, It would appear that study only depleted macrophages 6 days after SNI, a potentially important difference from the experimental design in the current study, which should be mentioned. Furthermore, the discussion may benefit from acknowledging there are other studies that reached a similar conclusion [Old EA et al. (2014), JCI 124(5): 2023, De Logu F et al. (2017), Nat. Commun. 8(1): 1887], and offering potential explanations as to why different groups are reaching different conclusions.

9. While it is true that the study cited by Peng et al (2015) showed that selective depletion of peripheral monocytes/macrophages spared DRG macrophages and did not abolish neuropathic pain, it is important to note that the phrase "... had no impact on neuropathic pain development" could cause readers to develop an incorrect perception of the paper's findings. Peng et al. write in their introduction:

"...Our results indicate that depletion of both resident microglia and peripheral monocytes completely prevented the development of neuropathic pain. However, either resident microglia or peripheral macrophages are critical for the initiation of neuropathic pain, suggesting that they act synergistically to promote the transition from acute to chronic pain after peripheral nerve injury."

Furthermore, though Peng et al. write that their clodronate liposome depletion protocol depleted circulating monocytes and spared those in the DRG, they did not assess monocyte/macrophage depletion, or lack thereof, in the sciatic nerve. Therefore, it is unclear whether effective depletion of resident/infiltrating monocytes from the sciatic nerve was occurring in their hands.

Response to Reviewers' comments:

Reviewer #1 (Remarks to the Author):

In this study, Yu et al. investigated the contribution of DRG macrophages to peripheral nerve injury-induced neuropathic pain. There are several novel findings in this study: (1) Using an interesting mouse line (MAFIA), the authors were able to selectively deplete peripheral macrophages without impacting microglia. The depletion of macrophages attenuated both initiation and maintenance of SNI-induced mechanical hypersensitivity. (2) Using local AP to kill macrophages at the injury site (but had no systemic effect), the authors found that DRG macrophages but not those at the nerve injury site are critical contributors to neuropathic pain. (3) Conditional deletion of CSF1 from sensory neurons prevents the injury-induced expansion of DRG macrophages. On the other hand, macrophage deletion attenuated SNI-induced BDNF expression in sensory. (4) Given the reported sexual dimorphism in the microglial contribution to pain filed, the authors examined the role of DRG macrophages in both male and female mice. They found that SNI-induced expansion of DRG macrophages in both male and female mice, and depletion of DRG macrophages reduced SNI-induced chronic pain in both male and female mice. Overall, the results are somewhat surprising, but very exciting. The experiments were well designed and nicely performed. The study will shed new light on the role of peripheral macrophages in neuropathic pain and thus may provide potential pain therapeutics. Some concerns need to be addressed:

Major concerns:

1. Since the systemic AP treatment also depletes most of the dendritic cells and other tissue macrophages, the transient pain relief may come from sites other than DRGs, for example, the nerve fiber between DRG and injury site, the skin where the nerve fiber terminals located. Although the authors titrated the AP dose, other side effects cannot be completely ruled out. Indeed, the authors observed the weight loss after systemic AP application (Fig S4F). Therefore, the authors may need to discuss the possible other cell ablation than DRG macrophages as well as the complication of side effects of whole-body macrophages depletion.

Response: *We appreciate the Reviewer's point and have added a sentence in the introductory paragraph in the Discussion that now reads:*

"Although we cannot rule out a possible contribution of the systemic treatment-induced depletion of dendritic cells in the skin, we favor the view that DRG macrophages are the major contributor."

With regard to the possible contribution of systemic side effects other than weight loss contributing to the mechanical hypersensitivity observed after macrophage depletion, after addressing the weight loss question, we have added the following sentence to the Results section:

"Moreover, because thermal pain thresholds did not differ in the AP-treated mice, we conclude that other systemic side effects were not major contributors to the anti-allodynic effect produced by macrophage depletion."

2. The results are quite exciting that macrophages deletion without microglia ablation can attenuate SNI-induced pain hypersensitivity for at least a week. Particularly, it is interesting that increased microglial number still persists when pain is attenuated (Fig 2). Three questions: (1) wondering whether SNI-induced spinal microglial proliferation (Gu et al., Cell Rep, 2016) might be altered after DRG macrophage deletions; (2) similarly, how about SNI-induced CSF1 expression in the DRGs (Guan et al., Nat Neurosci, 2016)? (3) and how about the activation of DRG satellite cells or spinal astrocytes after macrophage deletion?

Response

(1). FACS analysis of spinal cord microglia 4 days after nerve injury (3 days of AP, followed by SNI on day 4 and examined 4 days later for FACS analysis) showed no difference between AP and VEH treated groups. We concluded that microglial proliferation is not affected when macrophages are depleted (Figure 1K).

(2). This is an interesting question. In fact, we now show that AP depletion did not affect nerve injury-induced de novo CSF1 expression (Figures S12A-B)

(3). We used GFAP immunostaining for spinal astrocytes (Figure S8) as well as a Connexin 43 marker for satellite cells in the DRG and found no effect of macrophage depletion (Figures S12C-D).

3. It is confusing to put together Fig 3A with the rest of figure, because Fig 3A characterizes behaviors after DRG macrophage deletion while the rest of Fig 3 describes macrophage deletion at injury sites. It might be okay to have Fig 1, 2, and Fig 3A in the same figure.

Response: We appreciate the concern and have regrouped the figures into Figure 1.

4. Since macrophages repopulate quickly after ablation (presumably less one week), it is not clear why the pain hypersensitivities came back long after macrophages restored. It is possible that the ablation time duration was not long enough to cover the critical time window for the neuropathic pain development. Wondering whether prolonged AP treatment (to cover SNI treatment day) may completely reverse the chronic pain development as shown by both microglia/macrophages ablation (Peng et al., Nat Neurosci, 2016)? Also, it would be good to add a more time points describing the time course of macrophage depletion and repopulation in DRGs.

Response: We appreciate the significance of this experiment and have added an experiment in which we extended the AP treatment from 3 to 4 days in order to cover the period in which the SNI was performed (i.e. on day 4). In Figure S9 we show that the prolonged AP treatment did not further delay development of the mechanical allodynia, and thus clearly did not prevent its appearance after about 9 days.

5. Along the similar line, the macrophage depletion and repopulation in DRGs should be shown to examine the role of macrophages in the maintenance of neuropathic pain (Fig. 4). The authors showed macrophage deletion can reverse the mechanical hypersensitivity between POD28 to POD37, but did not correlate the restoration of pain with macrophage repopulation.

Response: We appreciate the information that could be gained from this experiment, however, this would require a very extended analysis, taking months to breed sufficient mice, not to mention the actual duration of the experiment. Given that we showed a correlation between macrophage repopulation and hypersensitivity at the early time points after nerve injury, we request that this experiment not be required for this revision.

6. For the source of macrophage expansion in DRGs, the authors showed some conflicting results here. If the authors wanted to claim a major source from infiltration, immunostaining and the cell densities for both resident and infiltrated macrophages should be performed in addition to the FACS data (Fig S7). After SNI, the resident macrophages in DRG seem still dominant, thus the proliferation could contribute more for the cell expansion. Consistent with this notion, the authors then showed that conditional deletion of CSF1 from sensory neurons abolished SNI-induced expansion of DRG macrophages (Fig. 4B). Unless CSF1 is able to induce CCR2+ macrophage proliferation (which is unlikely as they do not express CSF1R), the results on the source macrophage expansion after SNI are very confusing.

Response: We respectfully disagree with the Reviewer. To our knowledge all macrophages express CSF1R and are thus in a position to respond to CSF1 that could originate from injured sensory neurons. With respect to the possible use of CCR2 expression to identify infiltrating

macrophages as the source of the increase after nerve injury, we have added the following paragraph to the Discussion section:

“We also addressed the possible source of the injury-induced macrophage expansion in the DRG. Although CCR2-expression has been associated with infiltrating macrophages, we found many CCR2⁺ resident macrophages in the DRG of uninjured mice. We conclude that CCR2-expression in DRG macrophages is not a suitable marker that can distinguish infiltrating from resident macrophages. Furthermore, using Ki67 to document proliferating cells, we found that the percentage of Ki67⁺CX3CR1⁺ macrophages in the axotomized DRG more than doubled. Although we cannot rule out a contribution from infiltration, we favor the view that macrophage expansion originates predominantly from resident DRG macrophages that proliferate after injury.”

Note also that the new Figure S1E illustrates coexpression of CCR2-RFP and CSF1R-GFP in macrophages.

7. The claim on IL1b dependent mechanism is relatively weak (Fig 5F) as DRG macrophages presumably produce a slew of proinflammatory cytokines after SNI. To strengthen the conclusion, a few additional experiments would be helpful: (1) A cytokine array would be good to screen the overall changes in a variety of cytokines/chemokines; (2) In addition to RT-PCR, it would be better to use immunostaining to show the increased expression of IL1b in DRG macrophages after SNI; (3) More importantly, it would be very convincing that IL1b local injection in DRGs can rescue the pain phenotype after DRG macrophage deletion.

Response: *We appreciate the value of performing a complete cytokine analysis, however, that was not the objective of the report, which is focused on the contribution of macrophages. We did highlight IL1b as an example of a cytokine that might contribute and as suggested by the reviewer, we now added in situ hybridization evidence for increased Il1b expression in macrophages after injury (Figures 5G-L). Importantly and in contrast to existing literature, we found no evidence for Il1b message in neurons, before or after nerve injury. Lastly, we consider a study of the effects of locally injecting Il1b in the DRG beyond the scope of our study. We are particularly concerned that such an experiment would injure the DRG, making interpretation of any results very difficult.*

Minor concerns:

1. Page 3, 2nd paragraph, “Chlodronate” should be “Clodronate”.

Response: *We have corrected the spelling.*

2. Page 4, it would be better to introduce that administration of AP in MAFIA mice selectively depletes macrophages but not microglia due to its impermeability to BBB.

Response: *We have modified the Introduction which now states the following with regard to AP treatment:*

“Based on reports that AP does not cross the BBB in MAFIA mice, this approach has been used to kill macrophages selectively.”

3. Fig 5, the authors might want to tone down BDNF results as BDNF is the mechanism for the pain phenotype in macrophage deletion mice. Fig 5E, there is no labeling for BDNF.

Response: *We are not clear as to the request of the Reviewer, but as we previously reported, BDNF deletion does not alter development of the mechanical allodynia following peripheral nerve injury.*

We consider the finding the AP treatment reduces BDNF expression, via an indirect action on macrophages of interest, regardless of the existing literature as to the contribution of BDNF to nerve injury induced pain processing.

4. The mechanical pain threshold in this study (around 0.5g) is much lower than usual, even their own study (e.g., around 1g in Guan et al., 2016). The author should explain the discrepancy.

Response: *In our hands, the thresholds that are determined can vary with age of the mice, however, the experimenter performing the von Frey test is probably the more critical factor. That is the reason why all analyses in any experiment must be performed by the same individual, which was the case in our studies. A different individual performed the behavioral testing in the Guan et al study.*

Not surprisingly, both ~ 0.5 gm and 1.0 gm of baseline mechanical threshold were displayed in a key paper from a respected lab (Mogil NN 2016, S Fig3A, S Fig 10A).

5. Fig S6C-D needs better descriptions in the figure legends and in the results.

Response: *Please note that Figures S6C-D have been moved and are now Figures S10 C-D, and we have amended the figure legends.*

Reviewer #2 (Remarks to the Author):

This paper investigated the role of macrophages in the DRG in neuropathic pain using transgenic mice that enable selective depletion of macrophages. They show that the initiation and maintenance of nerve injury-induced mechanical hypersensitivity are reduced by depleting macrophages in the DRG, but not at the site of injury. Nerve injury increases the number of DRG macrophages whose source could be circulating monocytes. The increase in DRG macrophages is suppressed by conditional knockout of CSF1 in DRG neurons. The authors also show no sexual dimorphism in the expansion of DRG macrophages after nerve injury and in the contribution to mechanical hypersensitivity, although fewer macrophages are induced in females. From these findings, the authors conclude that macrophages in the DRG, but not injured site, contribute to both the initiation and maintenance of neuropathic pain. While this is an interesting paper providing evidence for the role of DRG macrophages, I have serious concerns about key aspects of the paper.

Major points

1. Using a technique enabling a selective depletion of macrophages at the site of injury, the authors clearly show that these cells have no contribution to nerve injury-induced mechanical hypersensitivity. Under this experimental condition, macrophages could be intact in the DRG, leading to one of the main conclusions of this paper that DRG macrophages are critical contributors to neuropathic pain. However, the local treatment of AP at the site of injury might also spare macrophages in the sciatic nerve that have previously been implicated in neuropathic pain. Thus, it remains unclear whether the effect of AP on neuropathic pain is dependent on macrophage depletion in the DRG. Therefore, to clearly show the role of macrophages in the DRG, the authors should examine neuropathic pain initiation and maintenance using mice whose DRG macrophages are selectively depleted by local application of AP into the DRG. This is essential to strengthen the conclusion of this paper.

Response: *We agree that this would be an interesting experiment, however, it is difficult to make DRG injections, and multiple DRG injections would be required, without compromising the tissue,*

not to mention the DRG itself. This is precisely the reason why we compared systemic vs nerve injury site injection of AP.

2. The mechanism for the expansion of macrophages in the axotomized DRG is also unclear. From the data in Figure S7A,B showing an increase in CCR2+CSF1R+ macrophages in the DRG, the authors conclude that the injury-induced expansion results from infiltrating circulating monocytes. However, it seems that the injury-induced increase in CCR2+ macrophages in the DRG was slight, and infiltrating macrophages/monocytes could not be a major population in total DRG macrophages. The authors need to include immunohistochemical data of CCR2+CSF1R+ macrophages in the DRG before and after nerve injury. Also, the percentage of GFP+ cells in the DRG of bone marrow chimeric mice after the injury should be counted. Furthermore, proliferation of macrophages should also be tested using proliferation markers such as BrdU.

Response: As noted above, we agree that CCR2 is not a reliable marker of infiltrating macrophages, and indeed now show that there is CCR2 expression in macrophages from uninjured animals (Figures S1C-E). We have therefore significantly tempered our hypothesis as to the source of the macrophage increase. However, by showing that there is Ki67 increase in the CX3CR1 macrophages, we favor the view that local expansion predominates. (Figure 1B).

We have added the following paragraph to the Results Section:

“With a view to determining the origin of the injury-induced macrophage expansion in the DRG, we monitored expression of the chemokine receptor CCR2, which reportedly marks infiltrating macrophages²³, in a double transgenic CCR2-RFP/CSF1R-GFP mouse. As expected, we observed significant numbers of CCR2+ macrophages at the peripheral nerve injury site, compared to the contralateral uninjured sciatic nerve (Figure S1A-B)., however, we also found many CCR2+ macrophages in the DRG of uninjured mice (Figure S1C-E), indicating that CCR2 is not a reliable marker of infiltrating macrophages. In a separate experiment, we costained the CX3CR1+ macrophages with a Ki67 antibody to mark proliferating cells. One day after nerve injury (POD1), FACS analysis showed that the percentage of Ki67+CX3CR1+ macrophages in the ipsilateral DRG did not differ from the uninjured contralateral DRG (Figure 1B). However, POD4, the percentage of Ki67+CX3CR1+ macrophages in the ipsilateral DRG more than doubled (Figure 1B, Figures S2A-B). Paralleling this result and consistent with previous immunocytochemical findings, FACS analysis showed that proliferating microglia (Ki67+CX3CR1+) in the lumbar cord on POD4 ipsilateral to the nerve injury increased by more than 2-fold compared to the contralateral side (Figure S2C). Together, we conclude that axotomy-induced macrophage expansion in the DRG involves local proliferation. Based on this result, although we cannot rule out infiltration, we favor the view that proliferation from resident macrophages predominates.”

We are somewhat puzzled by the request that we count the percentage of GFP+ cells in the DRG of bone marrow chimeric mice. In fact, we found that 100% replacement of macrophages in the DRG after bone marrow transplant. i.e. every cell was GRP positive.

3. The authors conclude that systemic AP treatment had no effect on spinal cord microglia. The number of microglia in the spinal cord of AP-treated mice without injury is not changed by AP (although it was slightly decreased), but it is more important to show that the AP treatment does not affect the nerve injury-induced microgliosis because this is a crucial step for activation of microglia and neuropathic pain. Thus, the authors should examine the number of spinal cord microglia and the expression of CX3CR1 mRNA at 4 days after nerve injury in vehicle and AP-treated mice.

Response: As suggested by the Reviewer, we provide data indicating that the number (increase) of microglia in the spinal cord after injury is not affected by AP treatment (Figure 1K).

4. The mechanism for the cellular interaction between macrophages and sensory neurons in the DRG is still unclear. The authors show upregulation of BDNF and IL-1b, but how do they interact

and contribute to macrophage expansion or CSF1 upregulation? Also, is CSF1 upregulation in axotomized DRG neurons changed by macrophage depletion?

Response: As noted above, the *de novo* induction of CSF1 in DRG neurons is not altered after macrophage depletion (Figure S12A-B). As to the BDNF, IL1b interaction, we strongly believe that this interesting question is outside of the scope of our analysis. The key point that we are highlighting is that although AP depletes macrophages, there are clearly consequences for the DRG neuron, which may or may not be relevant to the pain phenotypes. We chose to examine BDNF as previous studies suggested that BDNF is expressed by microglia (and by extension macrophages). In fact, as we now show, there is no macrophage expression of BDNF, however, the macrophage can influence neuronal expression of BDNF. The BDNF result stands in contrast to other genes that we examined, for example NPY and galanin, both of which are induced by nerve injury in neurons, but in contrast to BDNF, their upregulation is not affected by macrophage depletion (Figure S13).

Minor points

1. In Figure 6B, DRG macrophage depletion by AP in females should be given.

Response: The data are now provided in Figure S14.

2. Two identical Figure S4 are included in this paper.

Response: The duplication (now Figure S6) has been corrected.

3. Peng et al. have recently demonstrated that depletion of circulating monocytes does not change the injury-induced increase in macrophages in the DRG (Nat Commun 7: 12029, 2016). An explanation for this discrepancy should be included in Discussion.

Response: In fact, we cited the Peng et al result in the Introduction (Ref 6) as it provided part of our rationale for developing an alternative approach to macrophage depletion. We believe that the difference reflects the more effective depletion of DRG macrophages produced in the MAFIA compared to the clodronate approach.

4. CCR2-knockout mice (CCR2-REF/REF mice), but not CCL2-knockout mice, should be used for examining the nerve injury-induced macrophage infiltration in the DRG.

Response: As noted above we no longer suggest that CCR2 marks infiltrating macrophages.

5. Satellite glia has been proposed to be a source of IL-1b in the DRG after nerve injury (Pain 158: 1666-1677, 2017). The immunohistochemical or in situ hybridization analysis should be performed to determine the IL-1b-expressing cells in the DRG.

Response: Indeed the previous studies using antibodies reported expression of IL1b in both satellite cells and neurons. However, as question of antibody specificity is always of concern, in the revised manuscript we performed in situ hybridization and now show that IL1b message is, in fact, only found in macrophages (Figures 5G-L)

Reviewer #3 (Remarks to the Author):

In this article, Yu and colleagues investigate the contribution of macrophage infiltration into the peripheral nervous system to both the initiation and persistence of neuropathic pain, using the

spared nerve injury (SNI) model in mice. Using a transgenic mouse line enabling inducible depletion of macrophages without targeting microglia, they demonstrate that macrophages accumulating in the DRG, but not those at the site of nerve injury, contribute to the initiation and maintenance of mechanical hypersensitivity/neuropathic pain. Further, they report that depletion of DRG macrophages reduced this mechanical hypersensitivity in male and female mice. They observe an interesting sexual dimorphism in the requirement of neuronal CSF1 for macrophage expansion, which was only observed in male mice. The origins of the macrophages contributing to the expansion in the DRG are also addressed.

The manuscript addresses several unresolved topics in the field of neuroinflammatory macrophage/microglia crosstalk in chronic pain. They employ existing methods in new ways to arrive at conclusions that will undoubtedly be of interest to others in the community. Statistical analyses appear to be appropriate throughout, and the level of experimental detail seems sufficient to allow researchers to reproduce the work. The manuscript is well-written and the figures well-presented.

1. Microglia express CSF1R, the promoter of which controls the MaFIA receptor expression. What happens to Iba1 expression following AP administration in the DRG, is that reduced also?

Response: *In Figures S7A-B, we demonstrate reduction of Iba1 expression in the DRG after AP treatment.*

2. In Figure S5A, there seems to be a large increase in the CD11b⁺ GFP^{lo} population following AP administration. Does this represent monocytes with low-level expression of CSF1R, or some other cell type perhaps? This could have a meaningful impact on the cell types infiltrating the PNS following AP administration and nerve injury.

Response: *This result is now presented in Figure S7C. We agree with the Reviewer that the CD11b⁺ GFP^{lo} population represents monocytic cells with lower expression of CSF1R, which is consistent with the previous report of Aikawa et al Nat. Med 2013. However, as noted in the response to Reviewers 1 and 2, because of the expression of CCR2 by DRG macrophages in uninjured mice, we now downplay the impact of infiltrating monocytic cells in the DRG.*

3. Have the authors employed some form of positive control to see if activation of sensory neurons is still able to elicit pain following AP depletion of monocytes/macrophages? Such a control would ensure that neurons are still capable of relaying a nociceptive stimulus after AP. This would lend credence to the idea that apparent prevention/loss of hypersensitivity is due primarily to loss of macrophages in the context of SNI, rather than some off-target effect e.g. lethargy/cachexia.

Response: *The Reviewer highlights an important control, and as noted above, the fact that heat withdrawal thresholds in the Hargreaves test were not affected indicates that nociceptive processing at the level of the afferents is not affected by AP treatment. This point is now included in the revised Discussion section.*

4. Figure 3. Why are the values in 3A depicted as absolute threshold, yet % of baseline is used for all subsequent assessments of mechanical sensitivity? Furthermore, can the authors explain why the percent reduction from baseline is only 50% or so in Figure 3E, yet it routinely appears to be much greater - 5-10% of baseline in other figures?

Response: *We included this result using absolute thresholds (now Figure 1L) as it best illustrates the important point that baseline thresholds between the VEH and AP treated mice, using the 3 dose regime, did not differ.*

As to the difference in the magnitude of the allodynia, that is, in fact, commonly observed in such experiments. The differences are likely more related to the individual who performed the analysis. We have no other explanation. Importantly all experiments were performed blind to drug or vehicle so we have confidence in the results.

5. The authors remark that local delivery of AP to the future site of injury did not prevent subsequent macrophage expansion in the DRG (Figure 3D). The right panel in figure 3D appears to show no significant difference between vehicle and AP treatment. I am guessing that the 'veh' value (normalized to 1) already reflects the macrophage expansion reported in earlier figures, and thus the reported lack of significant difference between the 'veh' and 'AP' values indicates DRG macrophage expansion was unaffected by prior AP?

Response: *The Reviewer is correct. The values in this Figure, now 2D are normalized, and show that there was no difference in the macrophage expansion in the DRG when we targeted the AP to the nerve injury site.*

6. The authors theorize that DRG macrophages are the main source of the observed increase in IL-1 beta, which facilitates sensory neuron sensitization in response to injury. Similar increases have also been reported in peripheral nerve lesions (e.g. Kleinschnitz C et al., J. Neuroimmunol 149 (2004) 77-83), the existence of which should be discussed.

Response: *We thank the Reviewer for pointing this out. Please note that based on our finding that Ilb in the axotomized DRG (or for that matter in uninjured DRG) is not found in neurons, but rather is in macrophages (Figure 5G-L). As a result, we can conclude that the Ilb recorded in the Kleinschnitz et al study must derive from local cells at the injury site, not from transport by nerves from the DRG. As the Kleinschnitz study used qPCR, they could not identify the source of the Ilb. For this reason, we do not believe that that finding is relevant to our in situ analysis in the DRG.*

7. This brings me to another point which should be discussed; could the authors speculate on whether there is an intrinsic difference in the function of macrophages that infiltrate the peripheral nerve versus the DRG which may underlie their results, or do they think that inflammatory mediators produced at the site of nerve injury cannot excite sensory afferents to the same extent as those adjacent to somata in the DRG?

Response: *We appreciate the suggestion of the Reveiwer, but have no evidence one way or the other as to the differential contribution of inflammatory mediators at the DRG and peripheral injury site. For this reason, we prefer not to speculate at this time.*

8. The authors present evidence that macrophages at the site of nerve injury do not contribute to neuropathic pain, contrary to the study by Shepherd et al., however, It would appear that study only depleted macrophages 6 days after SNI, a potentially important difference from the experimental design in the current study, which should be mentioned. Furthermore, the discussion may benefit from acknowledging there are other studies that reached a similar conclusion [Old EA et al. (2014), JCI 124(5): 2023, De Logu F et al. (2017), Nat. Commun. 8(1): 1887], and offering potential explanations as to why different groups are reaching different conclusions.

Response: *We are aware of these reports and now cite them in the Revision. Specifically in the Introduction we write:*

“Other studies demonstrated clodronate killing of macrophages in injured peripheral nerves, and concluded that these macrophages were critical to the neuropathic pain development. However, as these studies did not examine the DRG, the contribution of DRG macrophages could not be ruled out.”

9. While it is true that the study cited by Peng et al (2015) showed that selective depletion of peripheral monocytes/macrophages spared DRG macrophages and did not abolish neuropathic

pain, it is important to note that the phrase "... had no impact on neuropathic pain development" could cause readers to develop an incorrect perception of the paper's findings. Peng et al. write in their introduction:

"...Our results indicate that depletion of both resident microglia and peripheral monocytes completely prevented the development of neuropathic pain. However, either resident microglia or peripheral macrophages are critical for the initiation of neuropathic pain, suggesting that they act synergistically to promote the transition from acute to chronic pain after peripheral nerve injury."

Furthermore, though Peng et al. write that their clodronate liposome depletion protocol depleted circulating monocytes and spared those in the DRG, they did not assess monocyte/macrophage depletion, or lack thereof, in the sciatic nerve. Therefore, it is unclear whether effective depletion of resident/infiltrating monocytes from the sciatic nerve was occurring in their hands.

Response: *We appreciate the Reviewer's concern. However, please note that the results and the title of Figure 4 in the Peng et al paper state: "Depletion of peripheral monocytes does not affect neuropathic pain development", which is consistent with what we wrote. Nevertheless, to address the Reviewer's concern we have modified the sentence, which now states that depletion of peripheral monocytes has "limited impact on neuropathic pain development".*

REVIEWERS' COMMENTS:

Reviewer #1 (Remarks to the Author):

The authors have addressed most of my concerns. Two minor points should be clarified:

1. I agree that resident macrophages express CSF1R. What we don't know is whether infiltrated macrophages (derived from monocytes) express CSF1R.
2. I understand a complete cytokine analysis is not the main objective. However, a better rationale should be introduced for the cherry-picking of IL1b and TNF.

Reviewer #2 (Remarks to the Author):

This resubmission has been significantly improved.

Reviewer #3 (Remarks to the Author):

Many thanks to the authors for addressing the queries and concerns raised in the first round of review. Several key experiments have now been added, which strengthen the conclusions made by the authors. In those instances where critiques have not been answered with additional/revised data, the reasons given are justifiable.

The modifications to the manuscript aid the reader in understanding the conclusions and placing them in context. I have no further suggestions or modifications.

Response to Reviewers' comments:

REVIEWERS' COMMENTS:

Reviewer #1 (Remarks to the Author):

The authors have addressed most of my concerns. Two minor points should be clarified:

1. I agree that resident macrophages express CSF1R. What we don't know is whether infiltrated macrophages (derived from monocytes) express CSF1R.

Response: We thank the Reviewer for highlighting this important emerging topic in macrophage ontogeny and agree that there may be a population of CSF1R negative monocytes. For example, CSF1R-independent monocytes were identified after injection of an anti-CSF1R antibody (Squarzoni P, et al., Cell Reports 2014).

However, as the Reviewer concluded in the initial comments, our observation that conditional deletion of CSF1 from DRG sensory neurons abolished SNI-induced expansion of DRG macrophages, which are CSF1R⁺, support our current conclusion that macrophage expansion “originates predominantly from resident DRG macrophages that proliferate after injury”.

2. I understand a complete cytokine analysis is not the main objective. However, a better rationale should be introduced for the cherry-picking of IL1b and TNF.

Response: Our rationale for focusing on IL1b and TNF instead of performing a complete inflammatory profiling was, in fact, stated in the Results. Specifically, we wrote that these genes “are reportedly expressed in both neuronal and non-neuronal cells of the DRG and have been previously implicated in neuropathic pain”. Moreover, populated macrophages after chronic injury were found to maintain high levels of IL-1 β and TNF- α expression (Paschalis EL, et al., PNAS 2018). For this reason we followed these two genes in the initial screening to test the hypothesis concerning cross-talk between macrophages and sensory neurons in the DRG.

Reviewer #2 (Remarks to the Author):

This resubmission has been significantly improved.

Reviewer #3 (Remarks to the Author):

Many thanks to the authors for addressing the queries and concerns raised in the first round of review. Several key experiments have now been added, which strengthen the conclusions made by the authors. In those instances where critiques have not been answered with additional/revised data, the reasons given are justifiable.

The modifications to the manuscript aid the reader in understanding the conclusions and placing them in context. I have no further suggestions or modifications.